# CocoRNA: Collective RNA Design with Cooperative Multi-agent Reinforcement Learning

## Abstract

Ribonucleic acid (RNA) plays a crucial role in various biological functions, and designing sequences that reliably fold into specified structures remains a significant challenge in computational biology. Existing methods often struggle with efficiency and scalability, as they require extensive search or optimization to tackle this complex combinatorial problem. In this paper, we propose CocoRNA, a collective RNA design method using cooperative multi-agent reinforcement learning, for the RNA secondary structure design problem. CocoRNA decomposes the RNA design task into multiple sub-tasks, which are assigned to multiple agents to solve collaboratively, alleviating the challenges of the curse of dimensionality as well as the issues of sparse and delayed rewards. By employing a centralized Critic network and leveraging global information during training, we promote cooperation among agents, enabling the distributed policies to cooperatively optimize the joint objective, thereby resulting in a high-quality collective RNA design policy. The trained model is capable of completing RNA secondary structure design with less time and fewer steps, without requiring further training or search on new tasks. We evaluate CocoRNA on the Rfam dataset and the Eterna100 benchmark. Experimental results demonstrate that CocoRNA outperforms existing algorithms in terms of design time and success rate, highlighting its practicality and effectiveness.

## 1 Introduction

RNA plays diverse roles in biological systems, including encoding proteins, regulating gene expression, and acting as a catalyst in biochemical reactions (Dykstra et al., 2022). Since the function of RNA is closely tied to its structure, designing RNA molecules that can fold into desired structures has become a critical problem. We focus on the RNA secondary structure design problem, which involves creating RNA sequences that can fold into a desired secondary structure. This is a complex combinatorial optimization problem and is computationally challenging due to the vastness of the sequence space and the intricate base-pairing interactions.

Most existing RNA design algorithms rely on online search or optimization techniques. Methods such as RNAinverse (Hofacker et al., 1994), RNA-SSD (Andronescu et al., 2004), INFO-RNA (Busch & Backofen, 2006), and antaRNA (Kleinkauf et al., 2015) use stochastic search strategies to find sequences that fold into the target structure. However, for longer sequences, the high dimensionality of the search space severely affects efficiency and scalability.

Some learning-based methods (Eastman et al., 2018; Runge et al., 2019) aim to reduce reliance on random search but still require extensive search or optimization times. However, existing methods also face the curse of dimensionality: the design space grows exponentially with sequence length, leading to extremely high computational costs. This limitation restricts the applicability of some methods; for instance, RNAinformer (Patil et al., 2024) is a generative model-based design method but is only suitable for short sequences.

Reinforcement learning (RL) is a promising approach to addressing biological sequence design problems. RL enables agents to learn decision-making policies through trial-and-error and exploration without relying on labeled data. Additionally, RL allows for flexible optimization guidance through the use of reward functions, which can be tailored to encourage desired design outcomes. Another potential advantage of RL-based methods is the ability to offload time-consuming online

optimization to offline training, enabling zero-shot learning on new tasks. However, RL-based methods face challenges in the context of RNA design. The vast design space inherent to biological sequences increases the difficulty of policy exploration and learning, adversely affecting policy performance. Moreover, longer sequences exacerbate the issues of delayed and sparse rewards (Riedmiller et al., 2018), which makes it difficult for RL agents to receive sufficient feedback. As a result, existing RL-based methods either require long search times (Eastman et al., 2018) or require online optimization for each target structure during the design phase (Runge et al., 2019). While these methods show promise, they do not fully exploit the potential of learning-based approaches to generalize across different target structures.

To address the above challenges, we propose a collective RNA design method based on multi-agent reinforcement learning (MARL). By decomposing the problem and learning cooperatively, we alleviate the curse of dimensionality and mitigate sparse rewards, thereby improving the efficiency and quality of RNA design. Our contributions can be summarized as follows:

- We formulate the RNA secondary structure design as a collective design problem, where multiple agents cooperate to design RNA sequences. By dividing the task among multiple agents—each focusing on a specific part of the sequence or structural elements—we effectively reduce the complexity faced by individual agents, enabling more efficient policy learning and exploration.

- We propose COCORNA, a MARL method based on the Centralized Training with Decentralized Execution (CTDE) framework to learn collective RNA design policies. Additionally, we introduce a search-augmented experience replay method to improve learning efficiency. Once trained, COCORNA can efficiently design RNA sequences within just one or a few episodes, requiring minimal time and steps without the need for further training or search on new tasks.

- We conduct experiments on the Rfam dataset and the Eterna100-v2 benchmark. The results demonstrate that COCORNA surpasses existing methods in both design time and success rate. Moreover, across both datasets, COCORNA solves more sequences within shorter time limits, demonstrating superior generalization performance.

## 2 RELATED WORK

### 2.1 RNA SECONDARY STRUCTURE DESIGN

**Optimization methods**. Intelligent optimization algorithms have made significant progress in solving the RNA secondary structure design problem (Churkin et al., 2018). Methods such as INFO-RNA (Busch & Backofen, 2006) combine dynamic programming with stochastic local search for RNA inverse folding. MODENA (Taneda, 2010) utilizes multi-objective genetic algorithms, and antaRNA (Kleinkauf et al., 2015) employs ant colony optimization techniques. These methods face major challenges when dealing with longer sequences, as the cost of performing stochastic search increases exponentially with sequence length. Additionally, evaluating designed RNA sequences requires repeatedly applying folding algorithms like ViennaRNA (Lorenz et al., 2011), which are based on dynamic programming and have computation times that grow significantly with sequence length.

**Learning-based methods**. Recently, learning-based methods have partially alleviated this problem. Reinforcement learning approaches (Eastman et al., 2018; Angermueller et al., 2019; Runge et al., 2019; 2024b) have been proposed for RNA design and other biological sequence design problems. For instance, Eastman et al. (2018) used RL to perform local search in the RNA sequence space. Runge et al. (2019) combined RL with neural architecture search to improve RNA design policies. Additionally, generative model-based methods like RNAinformer (Patil et al., 2024) have been developed. While RNAinformer show promise, it require substantial computational resources and is thus only applicable to sequences shorter than 100 nucleotides.

Unlike traditional optimization or search-based methods, RL has the potential to enable agents to adaptively make decisions without additional training or optimization, thereby achieving few-shot or even zero-shot design when encountering new instances. However, previous RL-based works have not truly achieved zero-shot design. For example, LEARNA (Runge et al., 2019) requires hundreds

or even thousands of learning episodes when facing new RNA structures. The vast search space of the RNA design problem increases the difficulty of policy exploration and improvement, making it challenging to obtain policies with strong generalization capabilities.

Our work addresses these challenges by introducing an MARL framework that decomposes the RNA design problem into sub-tasks handled by multiple agents. Similarly, RNA-SSD (Andronescu et al., 2004) reduces the size of the search space by splitting the full sequence into shorter subsequences and optimizing each subsequence independently. However, combining solutions to local problems often does not yield a solution to the global problem, and RNA-SSD is unable to incorporate global knowledge. In contrast, our method fully leverages global information, promoting cooperation among multiple agents and guiding joint optimization. Another work that adopts a similar idea is GAMEOPT (Bal et al., 2023), which establishes a game between different optimization variables to decouple the combinatorial decision space into individual decision sets. Although both methods are based on problem decomposition, our approach fundamentally differs from GAMEOPT. Our goal is to train a set of policies that can cooperatively make dynamic decisions to complete the design task, rather than searching for an optimal solution consisting of variables.

### 2.2 MULTI-AGENT REINFORCEMENT LEARNING

MARL methods address decision-making problems involving multiple agents by employing RL techniques (Gronauer & Diepold, 2022; Oroojlooy & Hajinezhad, 2023). In MARL, agents learn to cooperate or compete by interacting with the environment and optimizing their policies based on a reward function that defines the task objectives. This approach allows agents to autonomously learn policies without human intervention. Algorithms such as COMA (Foerster et al., 2018), QMIX (Rashid et al., 2018) and MAPPO (Yu et al., 2022) have demonstrated significant potential on complex MARL benchmarks like the StarCraft Multi-Agent Challenge (Samvelyan et al., 2019; Ellis et al., 2024). Recently, MARL has been widely applied to various real-world multi-agent tasks, including multi-robot systems (Chen et al., 2017; Willemsen et al., 2021; Paul et al., 2022), production scheduling (Johnson et al., 2022), and autonomous driving (Shalev-Shwartz et al., 2016; Candela et al., 2022).

However, to the best of our knowledge, no prior work has modeled biological sequence design as a multi-agent decision-making problem and applied MARL to solve it. Our work is the first to explore the potential of MARL in RNA sequence design, leveraging the cooperative capabilities of multiple agents to tackle the high complexity of the RNA design problem.

## 3 PRELIMINARIES

### 3.1 RNA DESIGN PROBLEM

We consider the RNA secondary structure design problem: given a target secondary structure, design an RNA sequence that most stably folds into that structure. RNA secondary structures are commonly represented using dot-bracket notation, where unpaired nucleotides are denoted by dots ( . ), and base pairs are represented by matching parentheses: an opening parenthesis ( ( ) indicates the 5' end of a base pair, and a closing parenthesis ( ) ) indicates the 3' end. For example, the notation ( ( ( . . . ) ) ) represents a hairpin loop structure with base pairs at the ends and unpaired nucleotides in the loop.

At each position in the RNA sequence, there are four possible nucleotides: adenine (A), uracil (U), guanine (G), and cytosine (C). Designing an RNA sequence that folds into a specific secondary structure is a complex combinatorial optimization problem. The size of the search space grows exponentially with the length of the RNA sequence. For a sequence of length $l$, there are $4^l$ possible nucleotide combinations, making exhaustive search computationally infeasible for large $l$.

### 3.2 DECENTRALIZED PARTIALLY OBSERVABLE MARKOV DECISION PROCESS

We model the fully cooperative multi-agent reinforcement learning problem as a Decentralized Partially Observable Markov Decision Process (Dec-POMDP) (Oliehoek et al., 2008). A Dec-POMDP is defined by the tuple $\langle \mathcal{N}, \mathcal{S}, \{\mathcal{O}_i\}, \{\mathcal{A}_i\}, P, R, \gamma \rangle$, where:

- $\mathcal{N} = \{1, 2, \ldots, n\}$ is the set of $n$ agents.

- $\mathcal{S}$ is the global state space.
- $\mathcal{O}_i$ is the local observation space of agent $i$, and the joint observation space is $\mathcal{O} = \mathcal{O}_1 \times \cdots \times \mathcal{O}_n$.
- $\mathcal{A}_i$ is the action space of agent $i$, and the joint action space is $\mathcal{A} = \mathcal{A}_1 \times \cdots \times \mathcal{A}_n$.
- $P : \mathcal{S} \times \mathcal{A} \times \mathcal{S} \to [0, 1]$ is the state transition probability function, where $P(s_{t+1} \mid s_t, \boldsymbol{a}_t)$ denotes the probability of transitioning to state $s_{t+1}$ given state $s_t$ and joint action $\boldsymbol{a}_t$.
- $R : \mathcal{S} \times \mathcal{A} \to [R_{\min}, R_{\max}]$ is the shared reward function.
- $\gamma \in [0, 1]$ is the discount factor.

At each time step $t$, the environment is in state $s_t \in \mathcal{S}$. Each agent $i \in \mathcal{N}$ receives a local observation $o_t^i \in \mathcal{O}_i$ and selects an action $a_t^i \in \mathcal{A}_i$ according to its policy $\pi^i(a_t^i \mid o_t^i)$. The joint action is denoted as $\boldsymbol{a}_t = (a_t^1, a_t^2, \ldots, a_t^n)$. The environment then transitions to a new state $s_{t+1}$ based on the transition probability $P(s_{t+1} \mid s_t, \boldsymbol{a}_t)$, and all agents receive the shared reward $R(s_t, \boldsymbol{a}_t)$. The goal is to learn the optimal joint policy $\boldsymbol{\pi} = (\pi^1, \pi^2, \ldots, \pi^n)$ that maximizes the expected discounted cumulative return:

$$G = \mathbb{E}\left[ \sum_{t=0}^{\infty} \gamma^t R(s_t, \boldsymbol{a}_t) \right]. \tag{1}$$

### 3.3 Centralized Training with Decentralized Execution

The Centralized Training with Decentralized Execution (CTDE) (Kraemer & Banerjee, 2016) is a paradigm in MARL that leverages the advantages of both centralized and decentralized approaches. During the training phase, agents utilize global information to evaluate and improve their local policies. In the execution phase, agents select actions based on their local observations and policy functions. CTDE mitigates the non-stationarity problem in multi-agent environments by allowing agents to consider the joint dynamics during training, promoting interaction and cooperation among agents. At the same time, decentralized execution based on local observations alleviates the curse of dimensionality and enhances system scalability, reducing the dependency on global information and communication capabilities.

**Actor-Critic Methods with CTDE**: In the standard Actor-Critic framework (Barto et al., 1983), the Critic network learns the value function to evaluate the current policy, while the Actor network learns the policy function and adjusts the policy parameters based on the Critic's estimation. The Actor-Critic architecture can be naturally extended to multi-agent systems under the CTDE paradigm, resulting in centralized Critics and decentralized Actors (Lowe et al., 2017; Iqbal & Sha, 2019). During training, the global information is used to train the centralized Critic networks, and the outputs of the Critic are used to improve each agent's Actor network. During execution, agents make decisions independently based on their own policies, relying solely on their local observations.

## 4 Multi-agent Reinforcement Learning for Collective RNA Design

In this section, we provide a comprehensive overview of the proposed COCORNA method, detailing its problem decomposition framework, algorithm architecture, reward function, and the search-augmented experience replay approach. The pseudocode for COCORNA can be found in Appendix A.

### 4.1 Problem Decomposition

In a Markov Decision Process (MDP), a policy is a mapping from states to actions:

$$\pi : \mathcal{S} \to \mathcal{A}, \tag{2}$$

where $\mathcal{S}$ is the state space and $\mathcal{A}$ is the action space. Typically, we desire the state to encapsulate as much relevant information as possible to inform decision-making. In the RNA design problem, the information influencing decisions includes: (1) the complete target RNA secondary structure; and (2) the current RNA sequence, since nucleotides at different positions are interdependent due to

base pairing constraints. Therefore, the complete state information should be a combination of the RNA sequence and structural information. Formally, the state space $\mathcal{S}$ can be defined as:

$$\mathcal{S} = \{(s_{\text{seq}}, s_{\text{struct}}) \mid s_{\text{seq}} \in \{A, U, G, C\}^l, \ s_{\text{struct}} \in \{\,.\,, (, )\,\}^l\}, \tag{3}$$

where $s_{\text{seq}}$ represents the nucleotide sequence, and $s_{\text{struct}}$ represents the target secondary structure in dot-bracket notation. For sequences of length $l$, the dimension of the state space is $|\mathcal{S}| = (4 \times 3)^l$, considering all possible combinations of sequences and structures. This exponential growth results in a high-dimensional policy space, significantly increasing the difficulty of policy learning and exploration.

We propose employing multi-agent reinforcement learning within the CTDE architecture, where the original problem is decomposed into multiple sub-problems and assigned to multiple agents to solve cooperatively. Each agent is responsible for making decisions at specific positions or substructures, reducing the dimensionality of individual agents' state and policy spaces. Specifically, we adopt two decomposition schemes:

- **Position-based decomposition**: We divide the entire RNA sequence into $n$ segments, assigning an individual agent to each segment. Each agent is responsible for deciding the nucleotide at its assigned segment.
- **Structure-type-based decomposition**: We assign each agent to a specific structural type based on the dot-bracket notation. Each agent designs nucleotides only at positions corresponding to its assigned structural type.

For a single agent, its input is a lower-dimensional local observation, which consists of a fragment of the RNA sequence and structure centered at the current design position. The observation space $\mathcal{O}$ for an agent can be formalized as:

$$\mathcal{O} = \left\{ (o_{\text{seq}}, o_{\text{struct}}) \,\middle|\, o_{\text{seq}} \in (\{A, U, G, C, \emptyset\})^m, \ o_{\text{struct}} \in (\{\,.\,, (, )\,, \emptyset\})^m \right\}, \tag{4}$$

where $m$ represents the length of the observation window (an odd integer), and $\emptyset$ denotes a placeholder symbol for positions outside the sequence boundaries. Specifically, for agent $i$, the observation at time $t$ is:

$$o_t^i = \left( s_{\text{seq}}^{[i-\kappa, \, i+\kappa]}, \ s_{\text{struct}}^{[i-\kappa, \, i+\kappa]} \right), \tag{5}$$

where $\kappa = (m-1)/2$, and $s_{\text{seq}}^{[i-\kappa, \, i+\kappa]}$ denotes the subsequence of nucleotides from position $i - \kappa$ to $i + \kappa$, similarly for $s_{\text{struct}}^{[i-\kappa, \, i+\kappa]}$. If $i - \kappa$ or $i + \kappa$ is outside the range $[1, n]$, we pad the observation with $\emptyset$ symbols. This local observation captures the immediate context around the agent's assigned position.

By decomposing the complex RNA design problem into smaller sub-problems and assigning them to multiple agents, we leverage the strengths of MARL to efficiently explore the vast search space. Each agent focuses on a specific part of the problem, either a sequence segment or a structural type, allowing for specialized policy learning. The centralized Critic ensures that while agents operate based on local observations, their policies are aligned towards the global objective, resulting in more effective and efficient RNA sequence design.

## 4.2 Algorithm Architecture

We adopt multi-agent proximal policy optimization (MAPPO) (Yu et al., 2022) as our base algorithm due to its reliable performance in various multi-agent reinforcement learning benchmarks. In our framework, each agent is equipped with an Actor network for decision-making and a Critic network for policy evaluation. Each agent has its own set of network parameters, allowing for independent learning and adaptation to its specific role within the environment.

**Actor Network**: The Actor network of each agent takes as input a local observation vector $o_t^i$, which captures relevant information at the agent's designated position. The Actor outputs a probability

distribution over the action space. We employ a discrete action space defined as:

$$\mathcal{A} = \{A,\ U,\ G,\ C\}, \tag{6}$$

where each action corresponds to selecting a nucleotide type to be placed at a specific position in the RNA sequence. The policy of agent $i$ is denoted as $\pi_{\theta^i}(a_t^i \mid o_t^i)$, where $\theta^i$ represents the parameters of the Actor network. During interaction with the environment, each agent samples an action $a_t^i$ from the probability distribution output by its Actor network and executes this action, which corresponds to assigning a nucleotide to the current position.

**Critic Network**: The Critic network takes the global state $s_t$ as input, which includes the current RNA sequence and secondary structure information. We use two convolutional neural network modules to process these two parts separately. The outputs of these two modules are then concatenated and fed into the subsequent layers of the Critic network to produce the value estimate $V(s_t)$ of the global state, representing the expected cumulative future rewards starting from state $s_t$ under the current joint policy $\boldsymbol{\pi}$:

$$V^{\boldsymbol{\pi}}(s_t) = \mathbb{E}_{\boldsymbol{\pi}}\left[\sum_{k=0}^{\infty} \gamma^k r_{t+k} \ \middle|\ s_t\right]. \tag{7}$$

The value estimates from the Critic network are used to compute the Temporal Difference (TD) error $\delta_t$ and the advantage function $A_t$ for policy updates. The TD error is calculated as:

$$\delta_t = r_t + \gamma V(s_{t+1}) - V(s_t). \tag{8}$$

We use Generalized Advantage Estimation (GAE) (Schulman et al., 2015) to compute the advantage function:

$$A_t = \sum_{k=0}^{\infty} (\gamma\lambda)^k \delta_{t+k}, \tag{9}$$

where $\lambda \in [0, 1]$. The advantage estimates are then used to update the Actor network parameters $\theta^i$ by maximizing the MAPPO objective for each agent:

$$\max_{\theta^i} \mathbb{E}\left[\min\left(\frac{\pi_{\theta^i}(a_t^i \mid o_t^i)}{\pi_{\theta^i_{\text{old}}}(a_t^i \mid o_t^i)} A_t,\ \text{clip}\left(\frac{\pi_{\theta^i}(a_t^i \mid o_t^i)}{\pi_{\theta^i_{\text{old}}}(a_t^i \mid o_t^i)}, 1-\epsilon, 1+\epsilon\right) A_t\right)\right], \tag{10}$$

where $\epsilon$ is a small positive constant that determines the clipping range.

In summary, the Critic network's value estimates are used to guide policy improvement. Since the centralized Critic evaluates the performance of the entire multi-agent system based on global information, the local policies of each agent are encouraged to take actions that contribute to the global objective.

## 4.3 REWARD FUNCTION

By modeling the RNA design problem as an MDP, we can analogize RNA design to a navigation task—not in the physical world, but within the RNA design space. The gap between the current design and the target can be measured using the Hamming distance of RNA secondary structures. Inspired by navigation problems, we design a reward function that facilitates policy exploration and optimization in the RNA design space.

Firstly, we use the Minimum Free Energy (MFE) folding algorithm from the ViennaRNA package (Lorenz et al., 2011) to predict the secondary structure that each RNA sequence will fold into. Then, the Hamming distance between the folded RNA structure $x_f$ and the target RNA structure $x_t$ is defined as:

$$H(x_f, x_t) = \sum_{i=1}^{l} \mathbf{1}(x_f^i \neq x_t^i), \tag{11}$$

where $\mathbf{1}(\cdot)$ is an indicator function that equals 1 when $x_f^i \neq x_t^i$ and 0 otherwise, and $l$ is the length of the RNA sequence.

The reward function consists of two components: the intermediate reward and the final reward. At each time step $t$, the multi-agent team selects actions, and the environment immediately returns an

intermediate reward, which represents the change in the normalized Hamming distance between consecutive steps. If the folded RNA structure perfectly matches the target structure, i.e., $H = 0$, a final reward is given, and the current episode ends. The reward function is:

$$R_t = \begin{cases} (H_{t-1} - H_t)/l, & \text{if } H_t > 0, \\ C, & \text{if } H_t = 0, \end{cases} \tag{12}$$

where $C$ is a positive constant. The intermediate reward encourages the agents to iteratively reduce the differences between the folded RNA structure and the target structure by minimizing the normalized Hamming distance at each step. Meanwhile, the final reward $C$ is set to a relatively large value to incentivize the agents to find an optimal solution and escape potential local optima.

### 4.4 SEARCH-AUGMENTED EXPERIENCE REPLAY

Due to the vastness of the RNA design space, learning-based methods may face a cold-start problem: initial random policies perform poorly, leading to a lack of high-quality experience data for effective learning. Inspired by the Hindsight Experience Replay (HER) (Andrychowicz et al., 2017), we introduce a Search-Augmented Experience Replay (SAER) method, using a limited amount of greedy search during the early stages of training to improve data quality.

Specifically, after an agent selects an action $a$ according to its policy $\pi$ and receives a reward $r$, we perform a local search to check if there exists a better action $a'$ that yields a higher reward $r'$. If such an action is found, we replace the original action with the better one and modify the experience tuple $(o, a, r)$ to $(o, a', r')$. By doing so, we enhance the quality of the experience data, which in turn improves learning efficiency.

This approach helps alleviate the cold-start problem by providing the agents with higher-quality training data in the early stages. To prevent the local greedy search from causing the policy to converge prematurely to suboptimal solutions, we gradually reduce and eventually remove this operation as training progresses.

It is important to note that the replay buffer used here differs from those in off-policy RL methods (Mnih et al., 2015; Lillicrap et al., 2016). In our method, experiences are collected over multiple steps using the current policy until a fixed horizon is reached. At that point, the buffer is used to perform multiple gradient-based updates to both the Actor and Critic networks. Once the updates are completed, the buffer is cleared, and new experiences are sampled based on the updated policy.

## 5 EXPERIMENTS

### 5.1 DATASETS

In our experiments, we use two datasets: the Rfam dataset (Runge et al., 2024a) and the Eterna100-v2 benchmark (Koodli et al., 2021). The Rfam dataset is constructed based on the Rfam database (Kalvari et al., 2021). From the Rfam dataset, we randomly select 60,000 RNA secondary structures as our training set. It is important to note that these data do not contain labels, as RL algorithms learn policies through autonomous exploration and do not require labeled data. Additionally, we randomly select 650 RNA secondary structures as a test set, which are not included in the training set. The Eterna100 benchmark is a well-known set of challenging RNA design problems derived from the Eterna online RNA design game. More details about the datasets can be found in Appendix C.

### 5.2 RESULTS

**Experimental Setup:** We employ position-based decomposition with $n = 4$ agents. For each RNA secondary structure, we divide it into four subsequences of approximately equal length, each assigned to one of the four agents. The maximum number of steps per episode is set to 400. If a valid RNA sequence that folds into the target structure is found within 400 steps, the design process for that episode terminates successfully.

We first perform training over a total of 25 million steps. After training, we use the learned design policy to design RNA sequences for the RNA secondary structures in the test set. For each RNA

Table 1: Comparison of RNA design methods on Rfam datasets and Eterna100-v2 benchmark.

| Method | Results [Solved / All test samples] | |
| --- | --- | --- |
| | Rfam | Eterna100-v2 |
| COCORNA | **636**/650 (**97.85**%) | **70**/100 |
| LEARNA-30s | 64/650 (9.85 %) | 24/100 |
| LEARNA-60s | 133/650 (20.46 %) | 34/100 |
| LEARNA-300s | 368/650 (56.62 %) | 49/100 |
| LEARNA-600s | 380/650 (58.46 %) | 50/100 |
| Meta-LEARNA-Adapt-30s | 386/650 (59.38 %) | 49/100 |
| Meta-LEARNA-Adapt-60s | 452/650 (69.54 %) | 53/100 |
| Meta-LEARNA-Adapt-300s | 565/650 (86.92 %) | 57/100 |
| Meta-LEARNA-Adapt-600s | 581/650 (89.38 %) | 60/100 |

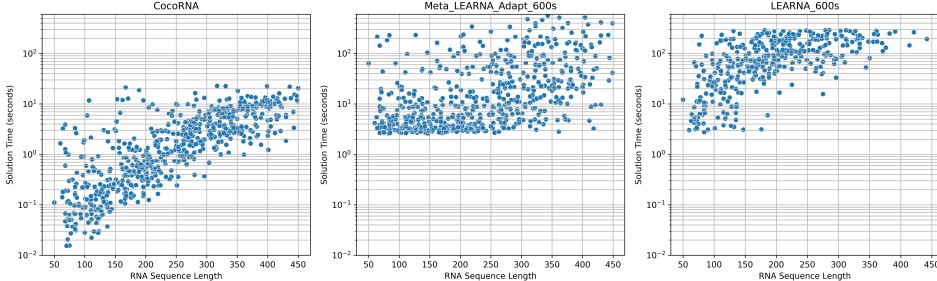

Figure 1: Distribution of solving times for COCORNA, LEARNA-600s, and Meta-LEARNA-Adapt-600s methods on successfully solved RNA sequences in the Rfam dataset.

structure, we allow a maximum of 15 retries, with a total time limit of 30 seconds. Details concerning the network architectures and hyperparameter settings are listed in Appendix D.

**Baselines:** We compare COCORNA primarily against the LEARNA algorithm (Runge et al., 2019), which, along with its variants, represents the state-of-the-art in RNA secondary structure design tasks. Compared to search-based methods such as antaRNA (Kleinkauf et al., 2015) and MCTS-RNA (Yang et al., 2017), as well as another RL-based approach (Eastman et al., 2018), LEARNA achieves faster speeds and higher success rates. Therefore, our comparisons focus on LEARNA and its variant, Meta-LEARNA-Adapt.

LEARNA uses RL to learn design policies and reduces reliance on random search, thereby improving the efficiency and quality of RNA design. However, LEARNA improves performance by conducting online optimization on specific RNA structures during the design phase, which increases resource and time costs. Without time constraints, the design phase can last hours or even days.

In contrast, COCORNA does not require random search or optimization during the design phase. Most RNA structures can be successfully designed within a single episode, and the design process usually takes less than a few seconds. To better compare algorithm efficiency, we test the performance of the baseline algorithms under time limits of 30s, 60s, 300s, and 600s.

**Comparison:** The results of the comparative study are presented in Table 1. COCORNA demonstrates clear advantages on both datasets. Under the same time limit of 30 seconds, COCORNA successfully solves 97.85% of RNA structures in the Rfam test set, substantially outperforming LEARNA (9.85%) and its meta-learning variant Meta-LEARNA-Adapt (59.38%). Even when the time limit for the baselines is relaxed by 20 times to 600 seconds, COCORNA still exhibits superior performance. On the Eterna100 benchmark, COCORNA also outperforms the baselines, successfully solving 70 out of 100 RNA structures.

Table 2: Distribution of RNA sequences by solving time intervals for different design methods on the Rfam dataset.

| Method | $t_{\text{solved}} < 1\text{s}$ | $1\text{s} \leq t_{\text{solved}} \leq 60\text{s}$ | $t_{\text{solved}} > 60\text{s}$ |
|---|---|---|---|
| COCORNA | 273 | 363 | 0 |
| LEARNA-600s | 0 | 147 | 233 |
| Meta-LEARNA-Adapt-600s | 0 | 474 | 107 |

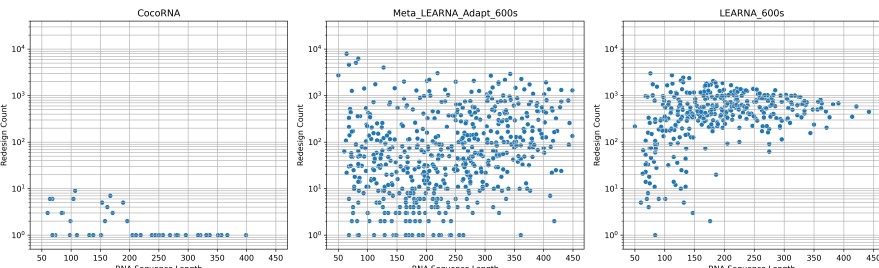

Figure 2: Distribution of redesign attempts required by COCORNA, LEARNA-600s, and Meta-LEARNA-Adapt-600s on the Rfam dataset.

Figure 1 shows the solving times for different RNA structures on the Rfam dataset and the relationship between solving time and sequence length for COCORNA, LEARNA-600s, and Meta-LEARNA-Adapt-600s. Complete test results, including other baselines, can be found in Appendix E. Table 2 presents the number of sequences whose solving times $t_{\text{solved}}$ fall within various time intervals for the three methods. The results indicate that COCORNA requires significantly less solving time, with approximately 42% of sequences successfully solved within 1 second. In contrast, neither of the baseline methods can solve any sequences within 1 second. Note that the solving time for our method increases with sequence length, primarily because most of the time is spent on the folding algorithm. While the inference time of our algorithm remains constant regardless of sequence length, longer sequences require more time to run dynamic programming-based folding algorithms (Lorenz et al., 2011).

Figure 2 shows the number of redesign attempts $n_{\text{iter}}$ required to design different RNA structures on the Rfam dataset using COCORNA, LEARNA-600s, and Meta-LEARNA-Adapt-600s. Table 3 presents the corresponding statistical information. For the majority of sequences (95.54%), our method succeeds within a single episode. In contrast, the two baseline methods require numerous iterative optimizations. Although Meta-LEARNA-Adapt performs meta-learning on the entire training set, only 3.38% of sequences can be solved in a single attempt.

Appendix E.1 discusses the impact of two different decomposition schemes: position-based and structure-type-based decomposition. Table 8 presents the experimental results, demonstrating that these two decomposition schemes have only a minimal effect on performance, suggesting that CO-CORNA is robust to the choice of decomposition strategy.

## 5.3 ABLATION STUDIES

To evaluate the effectiveness of each component in our proposed method, we conduct two ablation experiments: a single-agent version and an ablated multi-agent version without the SAER method. Figure 3 shows the results of these two ablation experiments. We plot the *Sequence Recovery Rate* (SRR) over the training process, which measures the ratio of correctly designed positions in the folded RNA sequence. The sequence recovery rate is defined as:

$$\text{SRR} = 1 - \frac{H(x_f, x_t)}{l}. \tag{13}$$

Table 3: Distribution of RNA sequences by redesign attempts $n_{\text{iter}}$ intervals for different methods on the Rfam dataset.

| Method | $n_{\text{iter}} = 1$ | $1 < n_{\text{iter}} \leq 15$ | $n_{\text{iter}} > 15$ |
|---|---|---|---|
| COCORNA | 621 | 29 | 0 |
| LEARNA-600s | 1 | 16 | 633 |
| Meta-LEARNA-Adapt-600s | 22 | 137 | 498 |

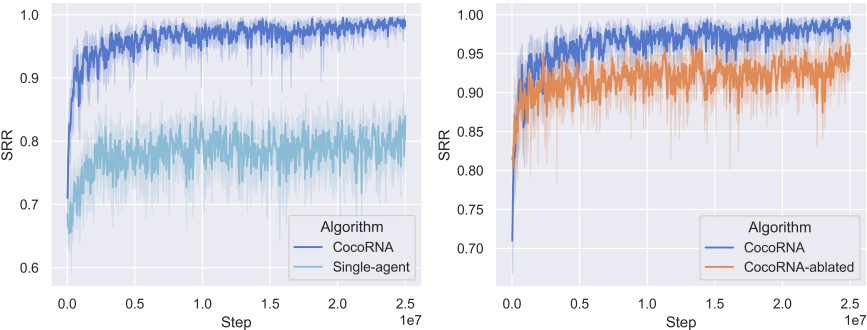

Figure 3: Results of ablation experiments on the Rfam dataset. Each experiment is performed over 6 independent runs with different random seeds. The shaded areas represent the standard deviation. **Left**: Comparison of sequence recovery rates during training between the single-agent version and our multi-agent COCORNA. **Right**: Effect of removing the search-augmented experience replay method.

As shown in the left plot of Figure 3, the training performance of the single-agent version is significantly worse compared to the multi-agent version, demonstrating the effectiveness of our proposed multi-agent cooperative design architecture. The right plot of Figure 3 shows that the ablated version without SAER converges more slowly and exhibits greater fluctuations during training, indicating that the introduction of SAER helps the agents learn more effectively by providing additional valuable experiences, thus stabilizing and accelerating the training process. Additional results of the SAER ablation study are provided in Appendix E.2.

Furthermore, we conducted a series of additional experiments to comprehensively evaluate the performance of COCORNA. These experiments include varying reward signal settings, agent size, shared policy parameters, dataset splitting, and training parameters. The detailed results and discussions for these experiments are presented in Appendix E.

## 6  CONCLUSION

In this paper, we proposed COCORNA, a collective RNA design method using cooperative multi-agent reinforcement learning to address the challenges in RNA secondary structure design. By formulating the RNA design task as a collective problem and decomposing it into multiple sub-tasks assigned to individual agents, the complexity faced by each agent was effectively reduced. Furthermore, the implementation of the CTDE framework combined with the introduction of the SAER method fostered multi-agent cooperation, enhanced policy exploration, and improved learning efficiency. Experiments conducted on the Rfam dataset and the Eterna100-v2 benchmark demonstrated both the effectiveness and efficiency of COCORNA.

We believe that the success of COCORNA showcases the potential of cooperative multi-agent reinforcement learning in biological sequence design tasks and complex combinatorial optimization problems. We anticipate that this method can be extended to more complex application scenarios, such as RNA tertiary structure design and protein design.

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

## A   PSEUDOCODE FOR COCORNA

Algorithm 1 presents the pseudocode for the proposed COCORNA method. Figure 4 illustrates the overall workflow of the algorithm. COCORNA leverages the MAPPO algorithm, an extension of the PPO method tailored for multi-agent environments. In our implementation of COCORNA, each agent is equipped with an Actor network responsible for selecting actions based on local observations, while the centralized Critic network evaluates the global state to provide value estimates for policy updates.

It is important to note that the replay buffer used here differs from those in off-policy RL methods (Mnih et al., 2015; Lillicrap et al., 2016). In our method, experiences are collected over multiple steps using the current policy until a fixed horizon is reached. At that point, the buffer is used to perform multiple gradient-based updates to both the Actor and Critic networks. Once the updates are completed, the buffer is cleared, and new experiences are sampled based on the updated policy.

---

**Algorithm 1** COCORNA: Cooperative Multi-Agent RNA Design

---

1: **Input:** Target RNA secondary structure dataset $\mathbb{X}$, maximum episodes $E$, maximum steps per episode $T$, horizon $H$, number of agents $n$
2: **Initialize** Actor networks $\pi_{\theta^i}$ and Critic network $V_\phi$ for each agent $i$
3: **Initialize** experience replay buffer $\mathcal{D}$
4: **for** episode = 1 to $E$ **do**
5:    Reset environment, randomly select a target RNA secondary structure $x_t$ from $\mathbb{X}$, and initialize RNA sequence $s_{\text{seq}}$
6:    Decompose the RNA design task among $n$ agents according to the chosen decomposition scheme
7:    **for** step $t = 1$ to $T$ **do**
8:       **for** each agent $i = 1$ to $n$ **in parallel do**
9:          Observe local observation $o_t^i$
10:          Select action $a_t^i \sim \pi_{\theta^i}(a_t^i \mid o_t^i)$
11:       **end for**
12:       Execute joint action $\boldsymbol{a}_t = (a_t^1, a_t^2, \ldots, a_t^n)$
13:       Update RNA sequence $s_{\text{seq}}$ with actions $\boldsymbol{a}_t$
14:       Predict folded structure $x_f$ using MFE folding algorithm
15:       Calculate Hamming distance $H_t = H(x_f, x_t)$
16:       Calculate reward $r_t$ based on $H_t$ and $H_{t-1}$
17:       Store transition $(\boldsymbol{o}_t, \boldsymbol{a}_t, r_t, \boldsymbol{o}_{t+1})$ in buffer $\mathcal{D}$
18:       **if** using SAER **then**
19:          Perform local search to find better actions $\boldsymbol{a}_t'$
20:          Update reward $r_t'$ based on improved Hamming distance $H_t'$
21:          Replace transition with $(\boldsymbol{o}_t, \boldsymbol{a}_t', r_t', \boldsymbol{o}_{t+1})$ in buffer $\mathcal{D}$
22:       **end if**
23:       **if** $H_t = 0$ or $t = T$ **then**
24:          Assign final reward $C$ to all agents
25:          **Break**
26:       **end if**
27:       **if** buffer size reaches horizon $H$ **then**
28:          Update Actor networks $\pi_{\theta^i}$ and Critic network $V_\phi$ using mini-batch updates from buffer $\mathcal{D}$ with MAPPO
29:          Clear experience buffer $\mathcal{D}$
30:       **end if**
31:    **end for**
32: **end for**

---

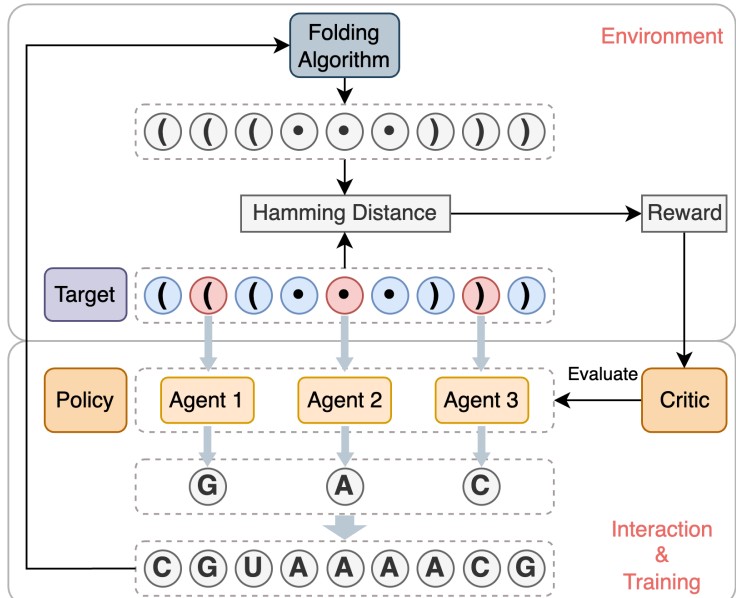

Figure 4: Overview of the COCORNA workflow.

## B  THEORETICAL ANALYSIS

### B.1  CONVERGENCE ANALYSIS

In this section, we provide a convergence analysis of the proposed algorithm. We demonstrate that, under certain conditions, the multi-agent actor-critic method used in COCORNA converges to a local maximum of the expected joint return.

In COCORNA, we employ a centralized Critic network to evaluate the joint policy. Specifically, we use the joint advantage function $A(s, \boldsymbol{a})$, defined as:

$$A(s, \boldsymbol{a}) = Q(s, \boldsymbol{a}) - V(s), \tag{14}$$

where $V(s)$ is the state value function. The joint policy gradient is then given by:

$$g = \mathbb{E}_{\boldsymbol{\pi}}\left[\sum_i \nabla_\theta \log \pi^i(a^i \mid o^i)\, A(s, \boldsymbol{a})\right] \tag{15}$$

$$= \mathbb{E}_{\boldsymbol{\pi}}\left[\sum_i \nabla_\theta \log \pi^i(a^i \mid o^i)\, Q(s, \boldsymbol{a})\right] - \mathbb{E}_{\boldsymbol{\pi}}\left[\sum_i \nabla_\theta \log \pi^i(a^i \mid o^i)\, V(s)\right], \tag{16}$$

where $\boldsymbol{\pi}$ denotes the joint policy, and $\theta$ represents the parameters of the Actor networks. The second term of (16) can be expanded as:

$$g_V = -\mathbb{E}_{\boldsymbol{\pi}}\left[\sum_i \nabla_\theta \log \pi^i(a^i \mid o^i)\, V(s)\right] \tag{17}$$

$$= -\sum_s d^{\boldsymbol{\pi}}(s) \sum_i \sum_{\boldsymbol{a}^{-i}} \boldsymbol{\pi}(\boldsymbol{a}^{-i} \mid o^{-i}) \sum_{a^i} \nabla_\theta \pi^i(a^i \mid o^i)\, V(s), \tag{18}$$

where $d^{\boldsymbol{\pi}}(s)$ denotes the discounted ergodic state distribution (Sutton et al., 1999), $\boldsymbol{a}^{-i}$ and $o^{-i}$ represent the joint actions and observations of all agents except agent $i$.

Due to the normalization property of the policy:

$$\sum_{a^i} \pi^i(a^i \mid o^i) = 1, \tag{19}$$

we have:

$$\sum_{a^i} \nabla_\theta \pi^i(a^i \mid o^i) = \nabla_\theta \sum_{a^i} \pi^i(a^i \mid o^i) = \nabla_\theta 1 = 0. \tag{20}$$

Therefore,

$$g_V = -\sum_s d^{\boldsymbol{\pi}}(s) \sum_i \sum_{\boldsymbol{a}^{-i}} \boldsymbol{\pi}(\boldsymbol{a}^{-i} \mid o^{-i}) V(s) \nabla_\theta 1 \tag{21}$$

$$= 0. \tag{22}$$

Thus, the policy gradient in (16) reduces to only the first term:

$$g = \mathbb{E}_{\boldsymbol{\pi}} \left[ \sum_i \nabla_\theta \log \pi^i(a^i \mid o^i) Q(s, \boldsymbol{a}) \right]. \tag{23}$$

Since the joint policy can be expressed as the product of individual agent policies:

$$\boldsymbol{\pi}(\boldsymbol{a} \mid s) = \prod_i \pi^i(a^i \mid o^i), \tag{24}$$

and using the logarithmic identity $\log \prod_i x_i = \sum_i \log x_i$, we can rewrite the policy gradient as:

$$g = \mathbb{E}_{\boldsymbol{\pi}} \left[ \sum_i \nabla_\theta \log \pi^i(a^i \mid o^i) Q(s, \boldsymbol{a}) \right] \tag{25}$$

$$= \mathbb{E}_{\boldsymbol{\pi}} \left[ \nabla_\theta \log \prod_i \pi^i(a^i \mid o^i) Q(s, \boldsymbol{a}) \right] \tag{26}$$

$$= \mathbb{E}_{\boldsymbol{\pi}} \left[ \nabla_\theta \log \boldsymbol{\pi}(\boldsymbol{a} \mid s) Q(s, \boldsymbol{a}) \right]. \tag{27}$$

This result shows that (27) is equivalent to the single-agent actor-critic policy gradient, where the multiple agents are considered as a single joint agent (or super-agent). Konda & Tsitsiklis (1999) proved the convergence of policy gradient algorithms under the condition that the policy $\boldsymbol{\pi}$ is differentiable and the algorithm parameters satisfy certain conditions.

Therefore, by following the gradient in (15), the multi-agent actor-critic algorithm converges to a local maximum of the expected joint return $J^{\boldsymbol{\pi}}$.

### B.2 LIMITATION AND DISCUSSION

In theory, gradient-based deep learning and reinforcement learning methods generally guarantee convergence to local optima rather than global optima due to the non-convex nature of the optimization landscape. However, in practice, the ability of reinforcement learning algorithms to go beyond local optima and approach global optima depends significantly on how well they balance exploration and exploitation.

Specifically, CocoRNA is built upon PPO, which includes an entropy bonus in the objective function, which explicitly encourages the policy to maintain high entropy. This prevents the policy from becoming too deterministic too quickly, avoiding premature convergence to suboptimal policies and promoting continued exploration.

In the context of the RNA design task, the multi-agent framework of CocoRNA further aids in overcoming local optima. Unlike single-agent methods that mutate one nucleotide at one step, our method allows multiple agents to act simultaneously on different parts of the RNA sequence. This introduces greater diversity and increases the exploration of the state space, thereby increasing the opportunities to discover better policies.

## C DATASETS DETAILS

We use the Rfam dataset (Runge et al., 2024a) and the Eterna100-v2 benchmark (Koodli et al., 2021) to train and evaluate our algorithm. Both datasets contain only RNA secondary structure

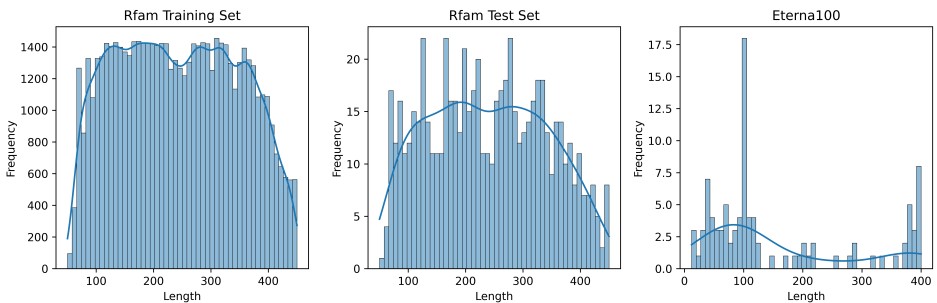

Figure 5: Length distributions of RNA secondary structures in different datasets.

Table 4: Statistics of the datasets.

| Dataset | Sample Size | Average Length | Max Length | Min Length |
|---|---|---|---|---|
| Rfam (Training Set) | 60000 | 243.73 | 450 | 50 |
| Rfam (Test Set) | 650 | 238.78 | 450 | 50 |
| Eterna100-v2 | 100 | 159.57 | 400 | 12 |

information without explicit sequence labels, as the task is to design sequences that fold into the provided structures.

The Rfam dataset is constructed by applying RNA folding algorithms to RNA sequences from the Rfam database (Kalvari et al., 2021). The structures in this dataset are computationally predicted using tools like the ViennaRNA package (Lorenz et al., 2011), which produces secondary structures based on minimum free energy (MFE) folding. We randomly sample 60,000 RNA secondary structures as the training set and an additional 650 structures as the test set.

The Eterna100-v2 benchmark consists of 100 challenging RNA design tasks derived from the Eterna citizen science project. These tasks represent a wide range of RNA secondary structures with varying lengths and complexities, providing a rigorous benchmark for evaluating RNA design algorithms.

Figure 5 shows the length distributions of RNA secondary structures in the Rfam training set, Rfam test set, and the Eterna100-v2 benchmark. Table 4 provides statistical information about the RNA structures in these datasets, including the sample size, average length, maximum length, and minimum length of the RNA secondary structures.

Table 5: Training hyperparameters.

| Hyperparameter | Value |
|---|---|
| Learning Rate | 0.00001 |
| Discount Factor ($\gamma$) | 0.99 |
| GAE Parameter ($\lambda$) | 0.95 |
| PPO Clip Range ($\epsilon$) | 0.2 |
| Entropy Coefficient | 0.02 |
| Horizon Length | 256 |
| Batch Size | 512 |
| Number of Epochs | 10 |
| Gradient Clipping | 0.5 |

Table 6: Network hyperparameters.

| Hyperparameter | Value |
|---|---|
| Actor Network Layers | [128, 64] |
| Critic Network Layers | [256, 64] |
| CNN Filter Size | [8, 4, 3] |
| CNN Number of Filters | [32, 64, 64] |
| Activation Function | ReLU |
| Optimizer | Adam |

# D ALGORITHM DETAILS AND HYPERPARAMETERS

This section provides a detailed description of the COCORNA algorithm, including network architectures, training details, and hyperparameter settings.

## D.1 NETWORK ARCHITECTURES

Each agent in the COCORNA framework is equipped with its own Actor and Critic networks. Although the Critic network leverages shared global information, the network parameters are not shared across agents. The Actor network takes local observations as input and outputs a discrete action (the nucleotide to be placed in the RNA sequence). The Critic network, on the other hand, processes the global state, which includes the full RNA sequence and secondary structure, and outputs a scalar value representing the estimated cumulative future reward for the entire system.

- **Actor Network:** The Actor network consists of several fully connected layers followed by a softmax output layer. Each agent's observation is passed through the network to compute a probability distribution over the action space $\{A, U, G, C\}$.

- **Critic Network:** The Critic network takes the entire RNA sequence and secondary structure as input. This input is processed through two separate CNN modules: one for the sequence and one for the structure. The outputs of these modules are concatenated and passed through a fully connected network to produce a value estimate for the entire system.

## D.2 TRAINING AND OPTIMIZATION

The training procedure involves collecting experiences over a fixed horizon, storing them in a replay buffer, and periodically updating the Actor and Critic networks using these stored experiences. The PPO algorithm is used to update the networks by maximizing the clipped objective, ensuring stable training. We also apply the Generalized Advantage Estimation (GAE) technique to compute the advantage function, which helps reduce variance during training.

Table 7: Additional hyperparameters.

| Parameter | Value |
|---|---|
| Maximum Steps per Episode (Training) | 350 |
| Maximum Steps per Episode (Testing) | 400 |
| Observation Radius | 30 |
| Local Observation Window Size | 61 |
| Global State Dimension | 900 |
| Total Training Steps | 25000000 |

## D.3 HYPERPARAMETER SETTINGS

Table 5 and Table 6 summarize the key hyperparameters used in training the COCORNA model, including training parameters and network parameters.

## D.4 OTHER SETTINGS

Table 7 presents additional parameter settings used in the COCORNA framework. During training, the maximum number of steps per episode is set to 350, while for testing, it is increased to 400. The training is conducted for a total of 25 million steps. Each agent has an observation radius of 30, resulting in a local observation window size of 61. Since the maximum RNA structure length in the dataset is 450, we standardize the global state dimension to 900 (450 for the sequence information and 450 for the structure information).

We use the ViennaRNA package (Lorenz et al., 2011) version 2.6.4 to compute the folded RNA structures during the design process.

Table 8: Performance of COCORNA with different problem decomposition schemes on Rfam and Eterna100-v2 datasets.

| Method | Results [Solved / All Test Samples] | |
| --- | --- | --- |
| | Rfam | Eterna100-v2 |
| COCORNA-PBD | 636/650 (97.85%) | 70/100 |
| COCORNA-SBD | 629/650 (96.77 %) | 68/100 |

Table 9: Performance of COCORNA and ablated versions on Rfam and Eterna100-v2 datasets.

| Method | Results [Solved / All Test Samples] | |
| --- | --- | --- |
| | Rfam | Eterna100-v2 |
| COCORNA | 636/650 (97.85%) | 70/100 |
| COCORNA-ablated | 603/650 (92.77 %) | 63/100 |
| Single-agent version | 344/650 (52.92 %) | 31/100 |

# E ADDITIONAL RESULTS

In this section, we present a series of experiments under various settings to comprehensively demonstrate the effectiveness of COCORNA.

## E.1 USING DIFFERENT DECOMPOSITION SCHEMES

Table 8 provides the testing results for different problem decomposition schemes used in CO-CORNA. We employ two decomposition methods: Position-based decomposition (COCORNA-PBD) and Structure-type-based decomposition (COCORNA-SBD). For Position-based decomposition, the complete RNA structure is divided into approximately equal segments, with each segment assigned to a different agent. For Structure-type-based decomposition, agents are assigned specific structural types based on the dot-bracket notation; for example, if Agent 1 is responsible for non-paired structures, it sequentially designs nucleotides at all non-paired positions within the structure.

In our experiments, the two decomposition methods only show minor performance differences, demonstrating the flexibility of our method concerning decomposition choices. For COCORNA-SBD, structure-type-based decomposition allows agents to focus more on specific structural elements, but the relative positional relationships between different agents may become completely

disrupted as the design process progresses, potentially causing slight adverse effects on the learning process. Although COCORNA-SBD performed slightly worse than COCORNA-PBD in testing, both methods significantly outperformed the single-agent approach.

## E.2    ABLATION STUDY ON SAER

Table 9 presents the testing results of two ablated versions of COCORNA. Removing the SAER method results in a slight performance decrease in COCORNA-ablated, demonstrating the effectiveness of the SAER approach. Additionally, as shown in the right plot of Figure 3, COCORNA-ablated continues to improve towards the end of the training process, suggesting that further training may yield better performance. The single-agent version shows significantly poorer performance. In fact, the single-agent version is similar to the LEARNA method but lacks architectural and hyperparameter optimizations and does not undergo additional training during the design phase.

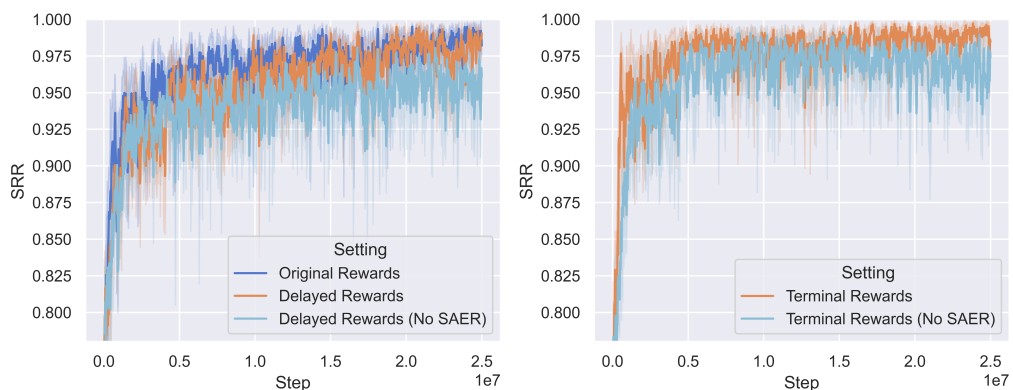

Figure 6: Learning curves under different reward settings on the Rfam dataset. Each experiment is performed over 6 independent runs with different random seeds. The shaded areas represent the standard deviation. **Left**: Learning curves using delayed reward (28). **Right**: Learning curves using terminal reward (29).

Table 10: Performance of COCORNA and COCORNA-ablated under different reward settings on the Rfam dataset.

| **Reward Setting / Method** | **Results** [Solved / All Test Samples] | |
| --- | --- | --- |
| Delayed Reward / COCORNA | 634/650 | (97.54%) |
| Delayed Reward / COCORNA-ablated | 621/650 | (95.54%) |
| Terminal Reward / COCORNA | 604/650 | (92.92%) |
| Terminal Reward / COCORNA-ablated | 570/650 | (87.69%) |

## E.3    ABLATION STUDY ON REWARD SIGNALS

In addition to the standard reward signal, we test two different delayed/sparse reward settings on the Rfam dataset.

- **Delayed Reward:** The reward signal is calculated every 10 steps instead of every step. The original reward function (12) is modified as follows:

$$R_t = \begin{cases} \frac{H_{t-10} - H_t}{l}, & \text{if } H_t > 0 \text{ and } t \bmod 10 = 0, \\ C, & \text{if } H_t = 0 \text{ and } t \bmod 10 = 0, \\ 0, & \text{otherwise.} \end{cases} \tag{28}$$

- **Terminal Reward Only:** The reward is given only at the end of an episode or upon successful design. The reward function is defined as:

$$R_t = \begin{cases} 1 - \frac{H_t}{l}, & \text{if } H_t \neq 0 \text{ and } t \text{ reaches maximum steps (episode ends)}, \\ C, & \text{if } H_t = 0, \\ 0, & \text{otherwise.} \end{cases} \tag{29}$$

We also test the impact of removing the SAER method under these two reward settings. The results are shown in Figure 6 and Table 10. We denote the ablated version without SAER as COCORNA-ablated. It can be observed that under the delayed reward setting, COCORNA experiences almost no performance loss. When only terminal rewards are available, the algorithm's performance decreases slightly due to the excessively sparse reward signal significantly increasing the difficulty of policy learning. Under both reward settings, COCORNA-ablated performs worse than COCORNA, demonstrating the effectiveness of the SAER method. Overall, this experiment shows that COCORNA is robust to different reward signals and performs well even under harsh reward settings. Notably, the delayed reward setting reduces the number of calls to the RNA folding algorithm to one-tenth of the original, significantly speeding up training.

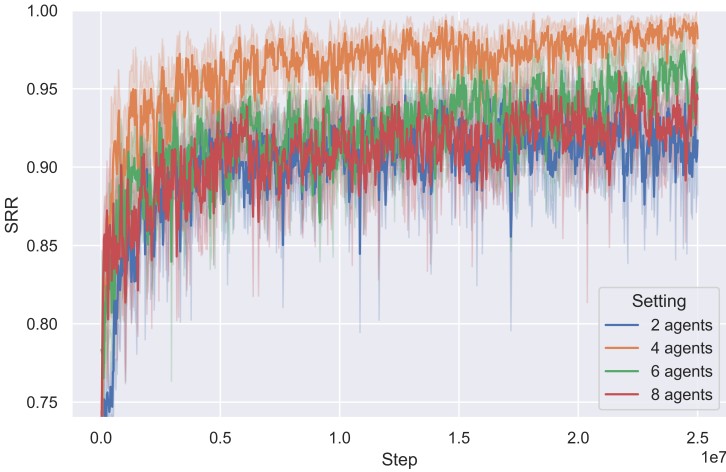

Figure 7: Learning curves of COCORNA under different agent size settings on the Rfam dataset. Each experiment is performed over 6 independent runs with different random seeds. The shaded areas represent the standard deviation.

Table 11: Performance of COCORNA under different agent size settings on the Rfam dataset.

| **Number of Agents** ($n$) | **Results** [Solved / All Test Samples] |
|:---:|:---:|
| $n = 2$ | 601/650  (92.46%) |
| $n = 4$ | 636/650  (97.85%) |
| $n = 6$ | 629/650  (96.77%) |
| $n = 8$ | 615/650  (94.62%) |

### E.4 EFFECT OF VARYING THE NUMBER OF AGENTS

In the main results presented earlier, we set the number of agents to $n = 4$. In this section, we adjust the parameter $n$ to examine the impact of different numbers of agents on the algorithm's performance. We experiment with four different settings: using 2, 4, 6, and 8 agents. Table 11

shows the performance of CoCoRNA under these different agent sizes on the Rfam dataset. The corresponding learning curves are presented in Figure 7.

From the results, we observe that using 4 or 6 agents achieves the best performance. When the number of agents is too small or too large, the performance tends to decrease. With too few agents, the benefits of multi-agent problem decomposition cannot be fully exploited, as each agent handles a larger portion of the task, leading to higher complexity per agent and potentially less efficient learning. On the other hand, with too many agents, the environment becomes more non-stationary from the perspective of each agent due to the increased interactions and dependencies among agents, making cooperation more challenging.

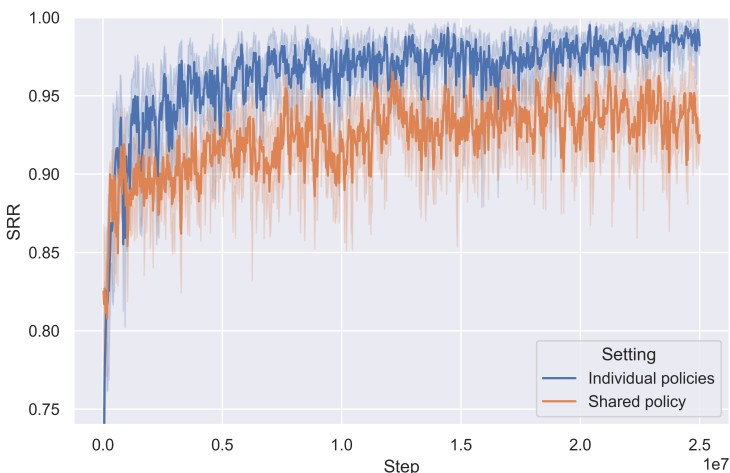

Figure 8: Learning curves of CoCoRNA-PBD and CoCoRNA-PBD-shared on the Rfam dataset. Each experiment is performed over 6 independent runs with different random seeds. The shaded areas represent the standard deviation.

Table 12: Performance of CoCoRNA with independent and shared policies on the Rfam dataset.

| Method | Results [Solved / All Test Samples] | |
|---|---|---|
| CoCoRNA-PBD | 636/650 | (97.85%) |
| CoCoRNA-PBD-shared | 622/650 | (95.69%) |
| CoCoRNA-SBD | 629/650 | (96.77%) |
| CoCoRNA-SBD-shared | 621/650 | (95.54%) |

### E.5  EFFECT OF SHARING POLICY PARAMETERS

We investigate the performance of CoCoRNA when agents share policy parameters under both decomposition methods. In these experiments, although different agents receive different observations, all agents share the same set of policy parameters. This approach is referred to as CoCoRNA-PBD-shared and CoCoRNA-SBD-shared for position-based decomposition and structure-type-based decomposition, respectively.

The learning curves and test results are shown in Figure 8 and Table 12, respectively. From the results, we observe that enforcing shared policy parameters among multiple agents leads to a slight decrease in algorithm performance compared to using independent policies, although the gap is relatively small.

These results potentially suggest that different agents learn distinct local policies that are specialized for their assigned sub-tasks. When agents share policy parameters, the policy learning process is constrained because all agents must use the same policy function, despite facing different sub-tasks or local environments. This can limit the agents' ability to adapt their policies to their specific roles in the RNA design task. In contrast, when using independent policies, each agent can tailor its policy parameters to its particular sub-task without interference from the data of other agents, providing more flexibility in policy learning.

Table 13: Performance of COCORNA on the re-split Rfam dataset, where the training and test sets are partitioned based on structural similarity.

| Method | Results [Solved / All Test Samples] | |
|---|---|---|
| COCORNA-PBD | 638/650 | (98.15%) |
| COCORNA-SBD | 628/650 | (96.62%) |

### E.6 DATASET SPLITTING

In the intersection of machine learning and biology, data leakage is a significant concern (Bernett et al., 2024). For supervised learning methods, if the test set contains data that are highly similar to those in the training set, it may not accurately assess the algorithm's performance on unseen data. However, since reinforcement learning is not a supervised learning method and the dataset does not contain explicit labels, data leakage is usually not an issue.

To further demonstrate COCORNA's generalization capability across different RNA structures, we re-split the dataset based on structural similarity rather than using a random split. Specifically, we first calculated the edit distance between each pair of RNA structures in the dataset, which consists of $65,000$ RNA structures. This resulted in a $65,000 \times 65,000$ similarity matrix that quantifies the pairwise differences between all RNA structures in the dataset. Next, we applied the MiniBatch K-Means clustering algorithm to partition the dataset into $100$ distinct clusters based on the calculated distances. We then selected one cluster at random to serve as the test set, consisting of $650$ RNA structures. We filtered the remaining data by removing RNA structures that had high similarity to the test set, and retained $60,000$ RNA structures as the final training set. The minimum edit distance between any structure in the test set and those in the training set is $111$, and the average edit distance is $188.42$.

Table 13 presents the results of training and testing on the re-split dataset. It can be observed that the differences in results across different datasets are minimal; different splitting methods do not significantly affect the algorithm's performance. Both COCORNA-PBD and COCORNA-SBD achieve high success rates, demonstrating the robust generalization ability of COCORNA to unseen RNA structures.

### E.7 TRAINING PARAMETERS

We observe that the algorithm is relatively sensitive to the learning rate compared to other hyper-parameters. In our main experiments, we use a learning rate of $1 \times 10^{-5}$, which is lower than the settings commonly used in RL algorithms. Figure 9 shows the learning curves when using higher learning rates. It can be seen that excessively high learning rates lead to unstable training and poor convergence.

In multi-agent environments, a high learning rate can exacerbate the non-stationarity of the environment. This issue arises because each agent's policy update affects the environment dynamics perceived by other agents. With higher learning rates, these changes become more abrupt, making it difficult for agents to adapt and learn stable policies. Consequently, the training process becomes highly unstable.

For other important hyperparameters, such as the discount factor $\gamma$, the GAE parameter $\lambda$, and the PPO clip range $\epsilon$, we use standard values commonly adopted in the literature. These settings are detailed in Appendix D.

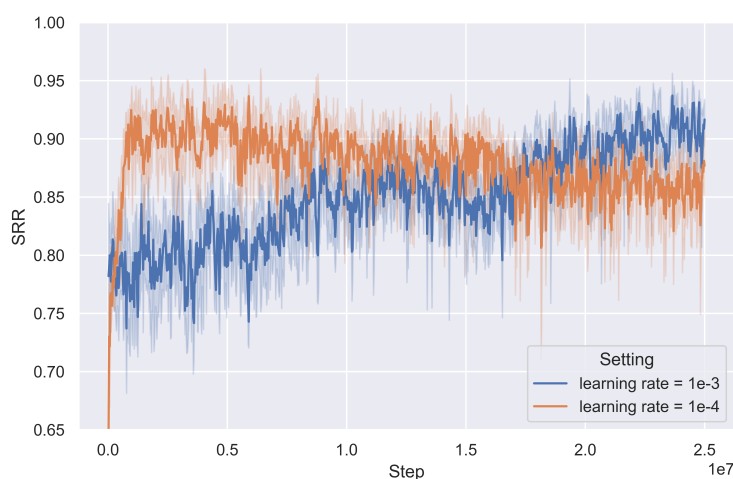

Figure 9: Learning curves of COCORNA using different learning rates.

### E.8 COMPARISON ON SEARCH-BASED METHOD

To further evaluate the performance of COCORNA, we conducted additional experiments comparing our method with antaRNA (Kleinkauf et al., 2015) on the Rfam dataset. antaRNA is an ant colony optimization-based RNA inverse design algorithm. Due to the time-consuming nature of its online search process, we limited the test set to 100 RNA structures.

We evaluated antaRNA under various computational time limits per RNA structure: 60 seconds, 600 seconds, and 1200 seconds. For the experiments with time limits of 60 seconds and 600 seconds, each RNA structure was designed only once. For the 1200-second time limit, we tested two settings: designing each RNA structure 5 times and 15 times, respectively, and selected the best result among the attempts. We report the average structural distance and the number of fully solved RNA structures (i.e., sequences that fold exactly into the target secondary structures). The results are presented in Table 14.

Table 14: Performance of antaRNA under different time limits on the Rfam dataset

|  | 60s (1 run) | 600s (1 run) | 1200s (5 runs) | 1200s (15 runs) |
|---|---|---|---|---|
| Average Structural Distance | 5.406 | 3.813 | 2.769 | 2.439 |
| Solved (Structural Distance = 0) | 12/100 | 14/100 | 22/100 | 22/100 |

As shown in Table 14, although antaRNA achieves relatively low average structural distances, the number of RNA structures it fully solves is significantly lower than that achieved by COCORNA, which attains a success rate of 97.85% (as reported in Table 1). This may be due to the presence of longer and more complex structures in the dataset, which affect optimization efficiency and make antaRNA more prone to getting stuck in local optima. Moreover, the maximum time limit of 1200 seconds per structure is substantially higher than the computational time required by COCORNA, highlighting the advantage of our method.

### E.9 OTHER RESULTS

Figure 10 shows the distribution of solving times for all baseline methods on the Rfam dataset, including only the sequences that were successfully solved. Similarly, Figure 11 presents the dis-

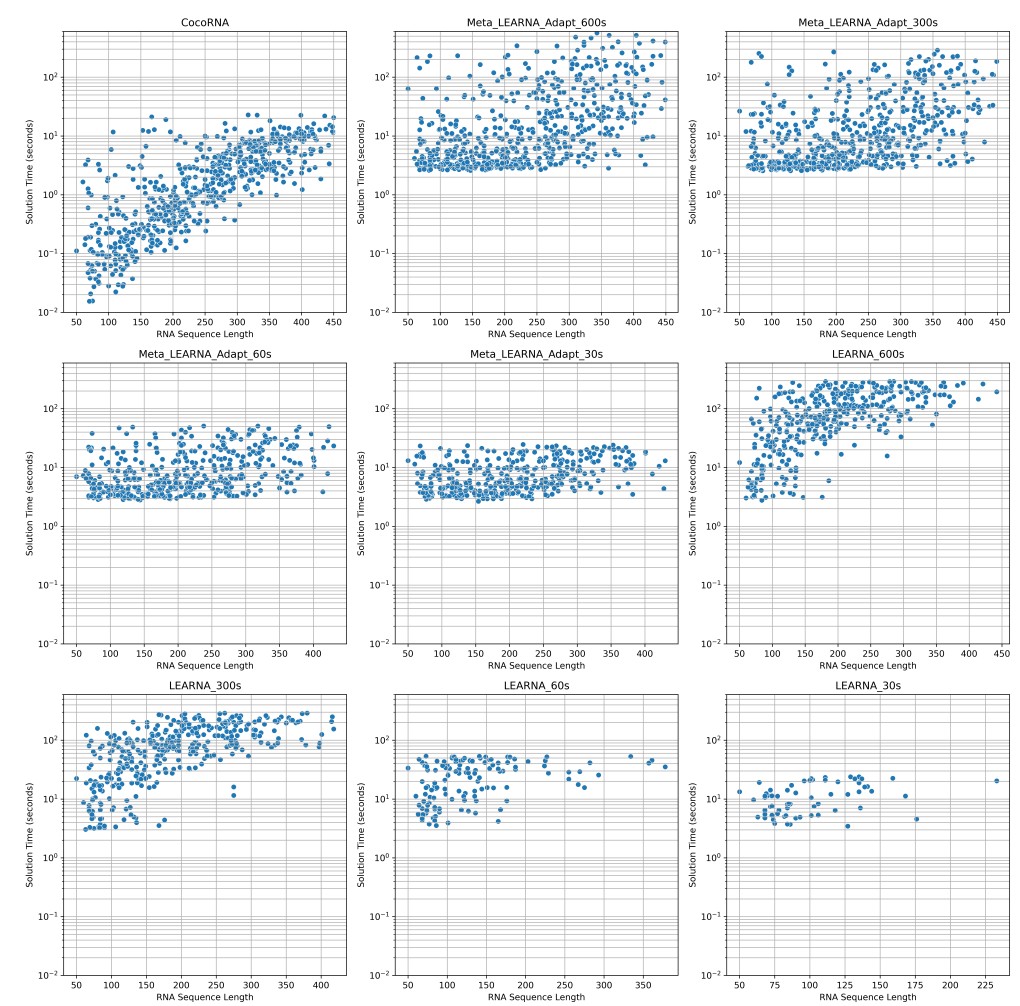

Figure 10: Distribution of solving times for different RNA design methods on the Rfam dataset.

tribution of redesign attempts (iterations) required by different RNA design methods on the Rfam dataset.

## F DISCUSSION

### F.1 GRAPH NEURAL NETWORKS

Many biological molecules are well-suited to be modeled as graphs due to their inherent structural properties. Transforming RNA structures into graph representations and utilizing Graph Neural Networks (GNNs) could potentially better capture the relationships between different nucleotides, enabling more effective policy learning. Graph-based representations naturally model the interactions and dependencies within RNA structures, which could enhance the agent's decision-making process.

However, employing more complex network architectures like GNNs also introduces higher computational and memory overheads, potentially reducing training efficiency. In our work, although we did not use GNNs, we addressed the issue of partial observations by employing CNNs within the centralized Critic to extract global structural information. This approach helps mitigate the limitations of local observations by providing a holistic view of the RNA structure, thereby supporting coordinated policy optimization without incurring excessive computational costs. Nevertheless, in-

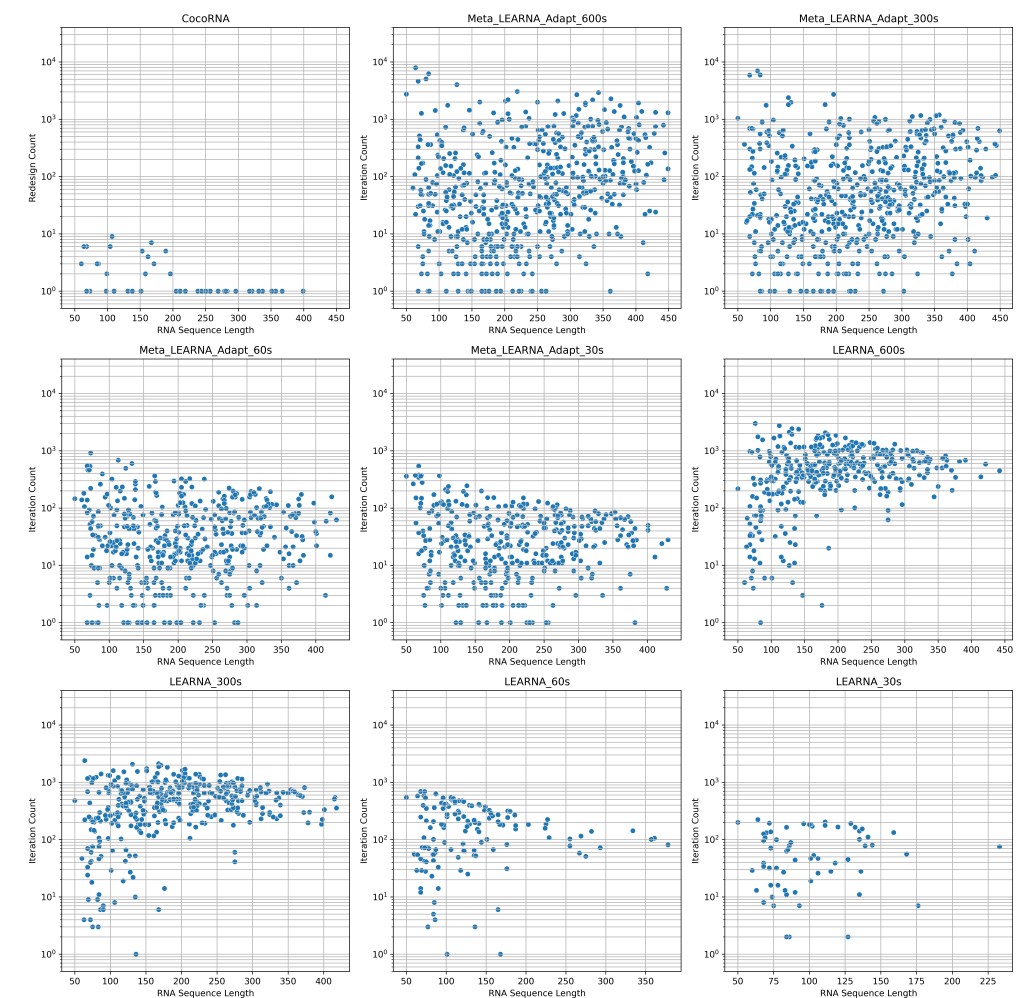

Figure 11: Distribution of redesign attempts (iterations) required by different RNA design methods on the Rfam dataset.

tegrating GNNs with reinforcement learning-based methodologies represents a promising direction for future research.

## F.2 LIMITATIONS

While COCORNA demonstrates promising results in RNA secondary structure design, there are several limitations.

First, it is non-trivial to design a accurate and reliable reward model given the complexity of biological systems. We may resort to large-scale pretrained models, e.g., RNA/protein language models. However, these models are often too large and thus are computationally expensive. One potential solution is to go with a model-based RL, where an explicit model of the environment is learned and used to predict future states and rewards more efficiently.

Another potential limitation is the decomposition method considered in the current CocoRNA. Given the high-dimensional nucleotide design space, it will be more promising to study an adaptive decomposition mechanism. This might be achieved by designing hierarchical policies, where high-level agents make decisions about task decomposition and low-level agents focus on specific sub-tasks.

Last but not least, this paper chooses the RNA inverse design problem as a proof-of-concept study. However, we believe CocoRNA or even MARL will be a potent method for solving such highly complex structured space problems. As part of our next step, we will explore other biological sequence design problems such as protein.

