# OpenReview forum: "CocoRNA: Collective RNA Design with Cooperative Multi-agent Reinforcement Learning"
_ICLR.cc/2025/Conference — Submitted to ICLR 2025_

### Official Review · Reviewer_rm6S · 2024-10-20

**Soundness:** 3
**Presentation:** 3
**Contribution:** 2
**Rating:** 5
**Confidence:** 3

**Summary:**

This paper presents a novel collective RNA design method based on cooperative MARL to solve the RNA secondary structure design problem. Empirical results demonstrate the outperformance of the proposed method.

**Strengths:**

The authors efficiently decompose the complex RNA design problem into mutiple sub-tasks. These tasks are allocated to cooperative agents to solve collaboratively. They introduce a search-augmented experience replay method to improve learning
efficiency, which improves the efficiency of RNA design. The proposed method significantly outperforms existing methods in terms of both design time and success rate.

**Weaknesses:**

1. This paper is more like an application of MARL in RNA design. The contributions to MARL method to solve the specific issues when applying MARL in RNA design should be stated clearly.
2. Besides the design of observations and reward functions, the authors should provide more explanations on how the agents cooperatively to sovle the RNA design task.
3. The authors do not discuss the limitations of the proposed method.

**Questions:**

See the weaknessnes above.

---

> ### Author Response · Authors · 2024-11-22
> **Response to Reviewer rm6S**
>
> We appreciate the reviewer’s valuable feedback. Please find our detailed responses below.
>
> **[W1]** **The novelty should be more clearly stated.**
>
> We elaborate on the novelty of our method from the three aspects:
>
> - We propose CocoRNA to solve the RNA inverse design problem, which is a highly complex combinatorial optimization task. We appreciate the reviewer's recognition that this is a difficult and significant problem. Most existing methods require performing online search for every new problem instance, which is computationally intensive and time-consuming. In contrast, CocoRNA is capable of achieving zero-shot design when encountering unseen RNA structures (please see **Figure 2** and **Table 3**).
> - We respectfully argue that CocoRNA is a simple application of MARL to solve a normal multi-agent problem. Instead, we explore the use of cooperative mechanisms in MARL to address large-scale combinatorial problems like RNA inverse design. Typically, multi-agent systems are considered more complex and unstable, which can hinder policy learning due to non-stationarity and coordination difficulties. However, our work demonstrates that MARL can be an effective approach to solving a single complex problem by decomposing it into sub-tasks handled by multiple agents that learn to cooperate. To the best of our knowledge, no prior work has modeled biological sequence design as a multi-agent decision-making problem or applied MARL to solve it. Our work is the first to explore the potential of MARL in biological sequence design, leveraging the cooperative capabilities of multiple agents to tackle the high complexity of the problem.
> - Methodologically, we propose a novel approach called Search-Augmented Experience Replay (SAER) to alleviate the cold-start problem caused by the high-dimensional decision space and to stabilize policy learning (please see results in **Figure 6**, **Table 9**, **Table 10**). We appreciate the reviewer's recognition of this method in the Strengths section. Additionally, we design a CNN-based centralized Critic network to effectively utilize global information, enabling better estimation of the global state-value function and facilitating cooperative policy improvement among agents.
>
> **[W2]** Besides the design of observations and reward functions, the authors should provide more explanations on how the agents cooperatively to sovle the RNA design task.
>
> We thank the reviewer’s comment and we address this from two aspects.
>
> - First, we have provided a detailed description of the algorithm architecture, including the composition of the Actor and Critic networks, in **Section 4.2** (line 263) and **Appendix D.1** (line 896).
> - Moreover, we have included a theoretical analysis in **Appendix B** (line 780) of the revised manuscript. The theoretical analysis supports our assertion that CocoRNA converges to a local optimum of the joint objective function. This enables multiple agents to cooperatively optimize the joint objective, rather than acting independently.
>
> **[W3] The authors do not discuss the limitations of the proposed method.**
>
> We discuss limitations as follows.
>
> - First, it is non-trivial to design a accurate and reliable reward model given the complexity of biological systems. We may resort to large-scale pretrained models (e.g., RNA/protein language models). However, these models are often too large and thus are computationally expensive. One potential solution is to go with a model-based RL, where an explicit model of the environment is learned and used to predict future states and rewards more efficiently.
> - Another potential limitation is the decomposition method considered in the current CocoRNA. Given the high-dimensional nucleotide design space, it will be more promising to study an adaptive decomposition mechanism. This might be achieved by designing hierarchical policies, where high-level agents make decisions about task decomposition and low-level agents focus on specific sub-tasks.
> - Last but not least, this paper chooses the RNA inverse design problem as a proof-of-concept study. However, we believe CocoRNA or even MARL will be a potent method for solving such highly complex structured space problems. As part of our next step, we will explore other biological sequence design problems such as protein.
>
> We have added the relevant discussion into the updated manuscript (please see line 1344, **Appendix F.2**).

---

> > ### Comment · Reviewer_rm6S · 2024-11-25
> >
> > I appreciate the authors' response. The novelty of the proposed method has beed stated clearly. I tend to keep my score.

---

> > > ### Author Response · Authors · 2024-11-26
> > > **Follow-up response to Reviewer rm6S**
> > >
> > > We sincerely thank the reviewer for engaging in this discussion and we are happy to know that the novelty concern has been addressed.
> > >
> > > However, since the current evaluation score is still 5, may we respectfully ask whether there are any further concerns at your side? We are keen on discussing with the reviewer. We believe such scientific debate will not only be important to minimize any confusion or misunderstanding in this peer-review. More importantly, if we do have  agreed on and identified some serious, un-changeable pitfalls, we will very much appreciate that and will fix them in due course. We believe this is exactly the beauty of the ICLR's rigor.
> > >
> > > Thank you very much!

---

### Official Review · Reviewer_nDf3 · 2024-10-27

**Soundness:** 2
**Presentation:** 3
**Contribution:** 2
**Rating:** 6
**Confidence:** 3

**Summary:**

This paper introduces a new approach for RNA secondary structure design by leveraging cooperative MARL. The proposed method, COCORNA, breaks down the RNA design task into multiple sub-tasks managed by individual agents. Through a centralized Critic and decentralized Actor architecture, COCORNA enables these agents to cooperate, aiming to address the combinatorial complexity of RNA sequence folding.

The model was trained on RNA design tasks using a novel Search-Augmented Experience Replay (SAER) mechanism, which improves initial learning efficiency. Experimental results on the Rfam and Eterna100-v2 datasets demonstrate that COCORNA outperforms existing methods in both design time and success rate, highlighting its potential for efficient and scalable RNA sequence design without further task-specific optimization. This study showcases COCORNA as a promising tool for addressing complex biological sequence design challenges through MARL.

**Strengths:**

1) **Significance of the Problem**: The paper tackles the complex challenge of RNA secondary structure design, which holds significant implications in fields like synthetic biology and gene regulation. By addressing the combinatorial nature of RNA design and the need for efficient, scalable methods, the authors provide a contribution to computational biology.

2) **Clear and Well-Structured Presentation**: The paper is well-organized, with each section logically progressing from the problem statement to methodology and experimental evaluation. The clear exposition of the multi-agent reinforcement learning framework, algorithmic details, and ablation studies makes it easy to understand both the innovation and the practical execution of the proposed approach.

3) **Robust Experimental Results**: The experimental results presented on the Rfam and Eterna100-v2 datasets are comprehensive and demonstrate COCORNA’s superior performance over existing methods in terms of success rate and design time. The ablation studies further substantiate the model's robustness, providing evidence that each component of the method contributes to the overall improvement.

4) **Intuitive and Effective Approach**: The proposed multi-agent reinforcement learning framework is well-suited to the complex, distributed nature of RNA design tasks. The decomposition of the design problem and the centralized training with decentralized execution approach provide a practical and computationally feasible solution. The inclusion of the SAER method to improve initial data quality and learning efficiency is a thoughtful addition that strengthens the model’s effectiveness in this challenging domain.

**Weaknesses:**

1) **Lack of Visual Aids for Method Explanation**: The paper lacks visual illustrations of the proposed method, which is a drawback given the complexity of multi-agent reinforcement learning and RNA secondary structure design. Effective diagrams and flowcharts could have greatly enhanced the readability and accessibility of the methodology, particularly for readers unfamiliar with MARL frameworks in biological sequence design. Including such visuals would improve the reader's ability to grasp the significance and innovative aspects of this work.

2) **Limited Novelty**: A significant concern lies in the limited novelty of the approach. The paper formulates RNA design as an MARL problem and primarily applies established MARL methods to solve it. The novelty of specific components, such as the reward function and the Search-Augmented Experience Replay (SAER) module, also seems limited. It would strengthen the work if the authors provided more innovative, task-specific adaptations or insights that build on existing methods in a unique way.

3) **Need for Deeper Conceptual Insights**: The paper would benefit from more in-depth conceptual insights into the unique challenges and opportunities specific to RNA design in the context of MARL. For instance, an analysis of how different decomposition approaches (e.g., position-based vs. structure-type-based) impact learning, or a discussion on task-specific challenges in RNA sequence alignment, would offer valuable perspectives. Such insights could highlight the authors’ deep understanding of the problem and provide a stronger foundation for the applicability and potential extensions of COCORNA.

**Questions:**

please see weaknesses.

---

> ### Author Response · Authors · 2024-11-22
> **Response to Reviewer nDf3**
>
> We sincerely appreciate the reviewer's recognition of the overall contribution of our work, and we address the raised concerns as follows.
>
> **[W1]** **The paper lacks visual illustrations of the proposed method.**
>
> As per requested by the reviewer, we have added a diagram to illustrate the workflow of CocoRNA (please see line 756, **Figure 4**).
>
> **[W2] The novelty should be more clearly stated.**
>
> We elaborate on the novelty of our method from the three aspects:
>
> - We propose CocoRNA to solve the RNA inverse design problem, which is a highly complex combinatorial optimization task. We appreciate the reviewer's recognition that this is a difficult and significant problem. Most existing methods require performing online search for every new problem instance, which is computationally intensive and time-consuming. In contrast, CocoRNA is capable of achieving zero-shot design when encountering unseen RNA structures (please see **Figure 2** and **Table 3**).
> - We respectfully argue that CocoRNA is a simple application of MARL to solve a normal multi-agent problem. Instead, we explore the use of cooperative mechanisms in MARL to address large-scale combinatorial problems like RNA inverse design. Typically, multi-agent systems are considered more complex and unstable, which can hinder policy learning due to non-stationarity and coordination difficulties. However, our work demonstrates that MARL can be an effective approach to solving a single complex problem by decomposing it into sub-tasks handled by multiple agents that learn to cooperate. To the best of our knowledge, no prior work has modeled biological sequence design as a multi-agent decision-making problem or applied MARL to solve it. Our work is the first to explore the potential of MARL in biological sequence design, leveraging the cooperative capabilities of multiple agents to tackle the high complexity of the problem.
> - Methodologically, we propose a novel approach called Search-Augmented Experience Replay (SAER) to alleviate the cold-start problem caused by the high-dimensional decision space and to stabilize policy learning (please see results in **Figure 6**, **Table 9**, **Table 10**). We appreciate the reviewer's recognition of this method in the Strengths section. Additionally, we design a CNN-based centralized Critic network to effectively utilize global information, enabling better estimation of the global state-value function and facilitating cooperative policy improvement among agents.
>
> **[W3]** **An analysis of how different decomposition approaches impact learning, or a discussion on task-specific challenges in RNA sequence alignment, would offer valuable perspectives.**
>
> We respectfully think this is a misunderstanding and we address your concerns as follows.
>
> - First, we have evaluated two different decomposition schemes: position-based and structure-type-based decomposition (please see **Table 8** in **Appendix E.1**). Further, we have added relevant explanations in the main text of the paper (see line 473). The experimental results demonstrate that these two decomposition schemes have only a marginal impact on the performance of CocoRNA, suggesting that our method is robust to the choice of decomposition strategy.
> - Regarding the complexity of the RNA design problem, we would like to provide further discussion.
>     - First, the RNA inverse design task is inherently complex as it requires the capture of intricate base-pairing relationships and potential interactions between nucleotides to achieve a desired structure. This involves both local and global dependencies within the RNA sequence, making the design process highly challenging.
>     - Second, the RNA inverse design is a high-dimensional combinatorial optimization problem. The search space for a sequence grows exponentially with its length. In our work, the search space for a single RNA structure can be as large as $4^{450}$, resulting in approximately $10^{271}$ possible combinations. Such a vast search space significantly increases the difficulty of designing algorithms capable of efficiently and effectively finding sequences that fold into the desired secondary structures.

---

> > ### Comment · Reviewer_nDf3 · 2024-11-26
> >
> > Thanks for your response. I think my concerns are mostly addressed, so I will improve my score to 6.

---

> > > ### Author Response · Authors · 2024-11-26
> > > **Follow-up response to Reviewer nDf3**
> > >
> > > We sincerely thank the reviewer for appreciating our justifications. If you have any other concerns, which are not fully or appropriately justified, we are keen on addressing them.

---

### Official Review · Reviewer_fWDL · 2024-11-03

**Soundness:** 2
**Presentation:** 2
**Contribution:** 3
**Rating:** 5
**Confidence:** 3

**Summary:**

The paper tackles the RNA secondary structure design problem proposing a novel approach using cooperative multi-agent RL. Multiple policies are jointly trained to design the sequence for parts of the RNA structure and a centralized critic is used to maximize the reward of the entire final sequence. The authors show that this methodology improves sample efficiency and computational efficiency while improving over other traditional RL-based baselines.

**Strengths:**

1.	A novel and interesting MARL-based methodology is proposed for RNA secondary structure design.
2.	A search-based augmented experience replay technique is proposed inspired by HER.
3.	The use of multiple policies improves sample efficiency issues and, given enough computational resources, also improves computational efficiency.

**Weaknesses:**

1.	The framework is still model-based and relies on repeatedly applying folding algorithms for reward calculation.
2.	Using the position-based decomposition, it is hard to justify having different local policies and not a shared policy. At least an ablation seems needed for that.
3.	Given the current manuscript, it is hard for readers to reproduce it, code is not available, there are some details that need additional information, and the random dataset splits might leak data.

**Questions:**

1.	It is not clear to the reviewer what is the output of the actor network. Is it the nucleotide type for 1 position or for all positions associated with that agent?
2.	Is code available for the proposed algorithm at an anonymous link?
3.	The reviewer suggests adding the part highlighting the difference from how the experience replay is applied from Appendix A to the manuscript.
4.	The authors mention two possible decompositions: (i) position-based; (ii) structure-based. In the experiments section, it seems that only position-based is mentioned. Are structure-based decomposition results also shown in the manuscript?
5.	The proposed method is compared with other RL-based baselines. It would be interesting to compare the proposed method to the other generative-based model baseline (Patil et al, 2024) having a small comparison regarding success rates and discussing which sequences can't be predicted by the generative-based model baseline because of their sequence length.
6.	From the reviewer's understanding, with n=4, four individual policies are trained. In this scenario, would not be the case to use four shared policies during training (with shared weights). It would be interesting to also have this ablation study.
7.	For the experiments, a random split of the datasets was performed. Similarly to protein-related tasks, a split based on hamming distances of structures to check the generalization capabilities of the policies would be desired in the reviewer’s opinion.
8.	It would be interesting to discuss the trade-offs between the proposed methodology and other generative-based alternatives such as graph neural networks using graph-based representations. As other parts of the structure might also affect the design of the sequence, having partial observations might not give enough information even with a centralized critic. For this, using a GNN architecture for decoding or a GNN-based policy for the RL methodology could alleviate this issue.

---

> ### Author Response · Authors · 2024-11-22
> **Response to Reviewer fWDL (Part 1)**
>
> We appreciate the reviewer’s useful and insightful comments. We have addressed each of the comments in the following responses.
>
> **[W1]. The framework is still model-based and relies on repeatedly applying folding algorithms for reward calculation.**
>
> We respectfully think this is a misunderstanding and we justify our stance from two aspects.
>
> - We clarify the differences between model-based and model-free RL as follows.
>     - In model-based RL, agents have access to an explicit model of the environment, which includes known state transition probabilities or reward functions. This model can be either handcrafted or learned from data, allowing the agent to plan actions by simulating possible future states.
>     - In contrast, model-free RL does not provide agents with prior knowledge. Instead, agents learn optimal policies solely based on observed interactions with the environment, receiving only observation information and reward signals without any explicit model.
>
>     Given these justifications, our proposed CocoRNA uses the RNA folding algorithm as the reward model while the agents do not possess any prior knowledge about the reward model or the state transition probabilities. Therefore, it is a model-free method.
>
> - We have conducted additional experiments on the reward signal, as presented in **Appendix E.3** (see **Table 10** and **Figure 6**). The results show that under the delayed reward setting, CocoRNA can reduce the number of calls to the RNA folding algorithm to one-tenth of the original frequency, with almost no performance loss. This demonstrates that our method does not heavily depend on frequent interactions with the reward model and can operate efficiently even with sparse reward signals.
>
> **[W2]. Using the position-based decomposition, it is hard to justify having different local policies and not a shared policy. At least an ablation seems needed for that.**
>
> We respectfully think this is a misunderstanding and we address the reviewer’s concerns from two aspects.
>
> - In our experiments, we have evaluated two different decomposition schemes: Position- and Structure-type-based decomposition (see **Table 8** in **Appendix E**). We have added relevant explanations in the main text of paper (Please see line 473).
> - Based on the reviewer's suggestions, we conducted ablation experiments on independent policies and shared policies. Please refer to our response to your **Q6**.
>
> **[W3]. Code is not available, there are some details that need additional information, and the random dataset splits might leak data.**
>
> We have uploaded our source code (see [code](https://openreview.net/attachment?id=4JZ56UVJYf&name=supplementary_material)). Other details concerned by the reviewer have been addressed to your **Q1**, **Q2**, **Q3**, **Q6**, and **Q7**.
>
> **[Q1] It is not clear to the reviewer what is the output of the actor network. Is it the nucleotide type for 1 position or for all positions associated with that agent?**
>
> We apologize for causing this confusion. As in Equation 6, the action space comprises the four nucleotide types {A, U, G, C}. Therefore, an Actor network outputs the nucleotide type for the current position at each time step. We have revised the relevant description in **Section 4.2** to clarify this point (Please see line 277).
>
> **[Q2] Is code available for the proposed algorithm at an anonymous link?**
>
> We have uploaded our source code (see [code](https://openreview.net/attachment?id=4JZ56UVJYf&name=supplementary_material)).
>
> **[Q3] The reviewer suggests adding the part highlighting the difference from how the experience replay is applied from Appendix A to the manuscript.**
>
> Thank you for your valuable suggestion. We have incorporated the explanation of experience replay from Appendix A into the main text of the manuscript (Please see **Section 4.4**, line 351).
>
> **[Q4] In the experiments section, it seems that only position-based is mentioned. Are structure-based decomposition results also shown in the manuscript?**
>
> Please kindly refer to our response to your **W2**.
>
> **[Q5] It would be interesting to compare the proposed method to the other generative-based model baseline (Patil et al, 2024) having a small comparison regarding success rates and discussing which sequences can't be predicted by the generative-based model baseline because of their sequence length.**
>
> We thank the reviewer for this suggestion. However, since the code of RNAinformer (Patil et al, 2024) is not available, we are unable to reproduce their results.
>
> As stated in their paper, RNAinformer has only been tested in RNA structures with lengths less than 100. Note that most RNA structures considered in our experiments have much longer lengths (see **Figure 5**). Because the complexity grows exponentially with the sequence length, the effectiveness of RNAinformer in problems with long sequences is questionable, at least not verifiable due to the lack of source code.

---

> ### Author Response · Authors · 2024-11-22
> **Response to Reviewer fWDL (Part 2)**
>
> **[Q6] From the reviewer's understanding, with n=4, four individual policies are trained. In this scenario, would not be the case to use four shared policies during training (with shared weights). It would be interesting to also have this ablation study.**
>
> We thank the reviewer for this suggestion. We have added the requested ablation study in this rebuttal (see **Appendix E.5**, **Figure 8**, **Table 12**). The results indicate that forcing multiple agents to share the same policy parameters leads to a slight decrease in algorithm performance. Using independent policies allows each agent to adapt its policy parameters to its unique role, providing greater flexibility in policy learning. Also, these results suggests that different agents learn distinct local policies that are tailored to their specific sub-tasks.
>
> **[Q7] For the experiments, a random split of the datasets was performed. Similarly to protein-related tasks, a split based on hamming distances of structures to check the generalization capabilities of the policies would be desired in the reviewer’s opinion.**
>
> We address the reviewer’s concern from following aspects.
>
> - First of all, we would like to clarify that data leakage problem is typically a concern for supervised learning, where datasets contain labeled data. In contrast, RL is not a supervised learning method, and the dataset used for RL does not involve labels. Instead, agents learn by interacting with the environment and receiving reward signals.
> - Nonetheless, we agree with the reviewer that using training and testing sets with low similarity is an effective way to better evaluate and showcase CocoRNA's generalization capabilities. Therefore, we have re-split the dataset to minimize the similarity. Specifically, we used the minimum edit distance to measure the similarity between two RNA structures, creating a similarity matrix and applying a clustering algorithm to categorize the RNA structures. This ensures that the similarity between the training and testing sets is as low as possible. Note that Hamming distance is not applicable in this context because it is used to compare structures of equal length, whereas our dataset consists of RNA structures with varying lengths.
> - As detailed in **Appendix E.6** (line 1174), we conducted additional experiments where the dataset was re-split based on structural similarity. **Table 13** presents the results of training and testing using this re-split dataset, which are consistent with the results in **Table 1**. This further demonstrates the generalization capability of CocoRNA.
>
> **[Q8] It would be interesting to discuss the trade-offs between the proposed methodology and other generative-based alternatives such as graph neural networks using graph-based representations.**
>
> Thank you for your insightful suggestion.
>
> - We agree with the reviewer’s perspective. Transforming RNA structures into graphs and utilizing GNNs could potentially better capture the relationships between different nucleotides, enabling more effective policy learning. Graph-based representations naturally model the interactions and dependencies within RNA structures, which could enhance the agent’s decision-making process.
> - However, employing more complex network architectures also introduces higher computational and memory overheads, potentially reducing training efficiency. In this work, although we did not use GNNs, we addressed the issue of partial observations by employing CNNs within the centralized Critic to extract global structural information. This approach helps mitigate the limitations of local observations by providing a holistic view of the RNA structure, thereby supporting coordinated policy optimization.
> - We also acknowledge that many biological molecules are well-suited to be modeled as graphs. Integrating GNNs with RL-based methodologies represents a promising direction for future research.
>
> We have added the relevant discussion into the updated manuscript (see line 1291, **Appendix F.1**).

---

> ### Author Response · Authors · 2024-11-26
> **Follow-up message to Reviewer fWDL**
>
> Dear Reviewer fWDL,
>
> Sorry for chasing the rebuttal. We thank you very much for your detailed and constructive suggestions and comments to our work. Would you please kindly let us know whether our responses really address your concerns? Or if it is the other way around, please feel free to let us know your further concerns and we are enthusiastic in this discussion.
>
> Thank you very much!

---

> > ### Comment · Reviewer_fWDL · 2024-11-27
> > **Rebuttal Feedback**
> >
> > Thank you for addressing my concerns and my comments.
> >
> > I have additional questions:
> >
> > **1. [Q6] Shared Policy:** In Table 12, it is interesting to observe that using the shared policy the decrease in algorithm performance is very small. When I mention a shared policy is related to my understanding that the local observations from different parts of the RNA would follow similar distributions. Is learning distinct local policy / sub-tasks crucial in this RNA secondary structure design problem?
> >
> > **2. [Q7] Dataset split:** Thank you for testing with a dataset based on structural similarity. I think it needs additional information regarding the split in Appendix E.6. How did you cluster the RNA structures? How low was the similarity applied for the distance metrics used? This information needs to be more clear for readers to reproduce these splits. Additionally, I tend to disagree with the sentence that in reinforcement learning data leakage / overfitting is usually not an issue. For example, if you train and test in the same environment (in this case, in similar structures) it is possible to just memorize the action sequence for that environment. Especially when translating to another domain like proteins, generalization is very important.
> >
> > **3. [Q8] GNNs:** Given my familiarity with inverse folding methods applied for proteins, it is very strange, in my opinion, the choice of baselines for this problem for RNA to be only RL-based. In my mind, methods like the one proposed in https://openreview.net/pdf?id=ByMEAHrgLB adapting the RNA secondary structure as node and edge features of a graph would also perform well. In this aspect, I share the same concerns with reviewer MViW [W9] regarding the baselines used and the use of a reliable world model when modeling the method.
> >
> > Given the efforts by authors to address my comments, I will increase my score to 5 but I still have main concerns regarding the methodology proposed in the manuscript.

---

> > > ### Author Response · Authors · 2024-11-30
> > > **Response to additional questions of Reviewer fWDL (Part 1)**
> > >
> > > **[Q6-2] Shared Policy:** **In Table 12, it is interesting to observe that using the shared policy the decrease in algorithm performance is very small. When I mention a shared policy is related to my understanding that the local observations from different parts of the RNA would follow similar distributions. Is learning distinct local policy / sub-tasks crucial in this RNA secondary structure design problem?**
> > >
> > > For the reviewer’s concern about [Shared policy … the local observations from different parts of the RNA would follow similar distributions …], we would like to justify that using a shared policy does not imply that all agents behave identically. In shared policy networks, agents are distinguished by incorporating agent-specific information (one-hot encoded agent IDs) into the policy input. This conditions the shared policy on the agent identity, allowing for individualized behaviors within a shared parameter framework. In MARL, sharing policy parameters among agents is a common practice that promotes learning efficiency and reduces the number of parameters. However, sharing parameters can limit flexibility to some extent, which may explain why, in **Table 12**, using independent policy networks yielded slightly better performance.
> > >
> > > For the reviewer’s concern about [… Is learning distinct local policy/sub-tasks crucial in this RNA secondary structure design problem?], our justifications are as follows.
> > >
> > > - Firstly, we respectfully argue that we do not assume that different parts of the RNA have different distributions. Moreover, we do not intend to deliberately obtain distinct local policies while we do appreciate that agents may indeed learn different local policies in practice.
> > > - As for the task decomposition in MARL, it does not require that sub-tasks have significant differences. In many MARL benchmarks, agents may be homogeneous and assigned similar or identical tasks. In CocoRNA, although agents may operate under similar policies, the division of the overall task into sub-tasks still provides the advantages discussed below.
> > >
> > > To further remove the reviewer’s concerns, we would like to clarify our rationale of employing multi-agent reinforcement learning (MARL) as the baseline for RNA inverse design problems.
> > >
> > > - Firstly, RNA inverse design is a large-scale combinatorial optimization problem. A single-agent approach would face high-dimensional state inputs, making policy exploration and improvement significantly more challenging due to the curse of dimensionality. By decomposing the task among multiple agents in CocoRNA, each agent operates on local observations and is responsible for a sub-task. This reduces the problem scale that each agent faces, simplifying the learning process. Additionally, a centralized Critic evaluates the global state to guide policy improvement, promoting cooperation among agents to achieve the overall design goal. Our experiments demonstrate that this multi-agent framework clearly outperforms the single-agent methods (see **Figure 3**).
> > > - The multi-agent mechanism in CocoRNA aids in promoting exploration of the vast combinatorial space in RNA design problem. Unlike single-agent methods that modify one nucleotide at a time, our approach allows multiple agents to act simultaneously on different parts of the RNA sequence. This increases the diversity and randomness of the data in the experience replay buffer, enriching the training experiences and encouraging broader exploration.
> > > - A significant challenge in RNA design is that only the final reward accurately reflects the quality of the designed sequence. While intermediate rewards can be provided, they may not fully capture the ultimate design objectives and can lead to suboptimal policies if overemphasized. In a single-agent framework, designing a sequence of length 400 would require 400 steps to complete, delaying the final reward and exacerbating the credit assignment problem. This delay makes accurate policy evaluation difficult and hinders effective learning. By employing multiple agents to collaboratively design the RNA sequence, we effectively shorten the time span to receive the final reward. This facilitates more accurate policy evaluation, improving the learning efficiency and overall performance.

---

> > > ### Author Response · Authors · 2024-12-02
> > > **Follow-up message to Reviewer fWDL**
> > >
> > > Dear Reviewer fWDL,
> > >
> > > Sorry for chasing this because the rebuttal window is very approaching the end. We sincerely appreciate the time and effort you have dedicated to reviewing our submission. We hope our responses have fully addressed your concerns. If you have any further questions or if there’s any additional information we can provide to assist in your evaluation, please do not hesitate to let us know.
> > >
> > > Thank you very much for your valuable time!

---

> ### Author Response · Authors · 2024-11-30
> **Response to additional questions of Reviewer fWDL (Part 2)**
>
> **[Q7-2] I think it needs additional information regarding the split in Appendix E.6. How did you cluster the RNA structures? How low was the similarity applied for the distance metrics used? This information needs to be more clear for readers to reproduce these splits.**
>
> We thank the reviewer for these further questions. We address the reviewers’ concerns from three aspects.
>
> - For the concern [How did you cluster the RNA structures?], we first calculated the edit distance between each pair of RNA structures in the dataset, which consists of $65,000$ RNA structures. This resulted in a $65,000 \times 65,000$ similarity matrix that quantifies the pairwise differences between all RNA structures in the dataset. Next, we applied the MiniBatch K-Means clustering algorithm to partition the dataset into $100$ distinct clusters based on the calculated distances. We then selected one cluster at random to serve as the test set, consisting of $650$ RNA structures.
> - For the concern [How low was the similarity applied for the distance metrics used?], we filtered the remaining data by removing RNA structures that had high similarity to the test set, and retained $60,000$ RNA structures as the final training set. The minimum edit distance between any structure in the test set and those in the training set is $111$, and the average edit distance is $188.42$.
> - As per requested by the reviewer, we have added a description of the dataset splitting in the revised manuscript (please see lines 1212-1221).
>
> **[Q8-2] In my mind, methods like [1] adapting the RNA secondary structure as node and edge features of a graph would also perform well. In this aspect, I share the same concerns with reviewer MViW [W9] regarding the baselines used and the use of a reliable world model when modeling the method.**
>
> For the concern about [… methods like [1] … a graph would also perform well], our justifications are from two aspects.
>
> - Firstly, we respectfully clarify that we do not intend to say graph neural networks as a generative model (e.g., [1] suggested by the reviewer) is not applicable for RNA inverse design. However, we respectfully think RNA secondary structure is not directly equivalent to protein 3D structure. Therefore, using the model in [1], which was designed for protein 3D structure, to a different domain would be a fair comparison. Instead, we think there should require substantial research (and we think this will be very interesting) to adapt the graph-based autoregressive model in [1] to our context. In other words, we think this deserves an individual paper, rather than simply using it as a baseline.
> - As we justified in our response to your **Q8**, we indeed appreciate the potential of using graph-based representation for tackling structured space like RNA or protein. Further, we think incorporating graph neural networks as the representation layer will be valuable for extending CocoRNA. Nevertheless, we respectfully think such synergy should not a simple plug-in and goes beyond the scope of this paper. In particular, given that the current CocoRNA has already achieved strong performance and efficiency, we would like to explore this as part of our future works.
>
> We will add a discussion about the above justifications in the Conclusion section in the camera-ready version.
>
> For the concern about [… regarding the baselines used …], we have added experiments on comparison with antaRNA [2] on the Rfam dataset (see **Appendix E.8**). Please kindly refer to the first bullet point of our response to **W9-2** of reviewer **MViW**.
>
> For the concern about [… use of a reliable world model …], we respectfully refer the reviewer to the second bullet point of our response to respectfully **W9-2** of reviewer **MViW**.
>
> > [1] Ingraham, J., Garg, V., Barzilay, R., & Jaakkola, T. (2019). Generative models for graph-based protein design. Advances in Neural Information Processing Systems, 32. (https://openreview.net/pdf?id=ByMEAHrgLB)

---

### Official Review · Reviewer_MViW · 2024-11-03

**Soundness:** 2
**Presentation:** 3
**Contribution:** 3
**Rating:** 3
**Confidence:** 4

**Summary:**

The paper addresses the challenge of efficient and scalable RNA secondary structure design. Designing RNA sequences that reliably fold into specified structures is difficult due to the complexity of the combinatorial search space. The paper proposes a collective RNA design approach called CocoRNA which uses cooperative multi-agent reinforcement learning. CocoRNA designs RNA sequences by decomposing the design task into subtasks assigned to multiple agents.

**Strengths:**

- The paper presents an efficient RNA secondary structure design method using a collective design approach, with empirical evaluations provided on two RNA design benchmark datasets.
- The paper demonstrates the potential of cooperative MARL approaches to RNA design tasks.
- The proposed approach addresses a real-world problem, validated through experiments on real-world datasets.

**Weaknesses:**

1. The main contribution of the proposed method is presented as designing RNA secondary structure as a "collective design" problem. However, the contribution is not novel as the recent work [1] already introduces such a collective design idea, that is, efficiently designing biological sequences using cooperative design framed as a cooperative game between players (here it is called agents), and [1] should be cited in the relevant work section. Based on this, it should be further clarified that what is the main source of benefit of using a MARL method, instead of performing collective design directly using combinatorial optimization as in [1]? What are the key differences and advantages of your method over [1]?

2. The paper lacks a discussion or a theoretical analysis of the convergence of the distributed policies to the global optimum. In the abstract and also in lines 126-129, it is stated that CocoRNA enables such a convergence, however, there is no guarantee or analysis that supports this claim. I would suggest that either a theoretical analysis or some clear discussion/intuition should be provided in the paper or the claims should be modified.

3. It would support the empirical performance of CocoRNA better if an analysis of the performance of CocoRNA under sparse and delayed rewards is presented. Instead of using the reward function in equation (12), how would the performance change under sparse reward (e.g., $R_t$ = {$0$ if $H_t > 0$; $C$ otherwise})?

4. While motivating CocoRNA in the Introduction, the authors state that RL-based methods do not exploit learning to generalize across different target structures (line 62). CocoRNA is stated to mitigate problems of (1) the curse of dimensionality and (2) sparse rewards (although not supported enough for (2)). However, the paper does not present how CocoRNA generalizes across (3) different target structures.

5. I think the related work section (2.1) should be restructured. There should be a section for RL-based methods that are used to design biological sequences such as [4,5]. These are potential SOTA baselines for CocoRNA. In addition, a discussion on why MARL is needed over these RL methods would clarify the contribution within RL & biological sequence design context. Instead of providing general MARL works in detail, this section would have provided the flow from RL to MARL within the problem context.

6. An ablation study on the grouping of players would be helpful in showing the effectiveness of CocoRNA regarding the interdependence of nucleotides at different positions. The paper presents only agent per position (n many agents) analysis. How does the performance of CocoRNA change with respect to the agent size?

7. Different than the point above, an ablation study considering the decomposition scheme such as position + structure assigned to an agent would have shown the proposed method's flexibility regarding decomposition choices. This is a more specific decomposition than only structure-based decomposition.

8. Regarding Section 4.4, in Hindsight Experience Replay (HER), additional goals are used to store additional episodes in the replay buffer to deal with sparse reward environments. Hence, the goal influences actions, but not the environment dynamics. In Search-augmented Experience Replay (SAER), how are the goals defined? How are additional goals for the replay sampled? It is not clear to relate SAER to HER and not provide clear explanations.  I think to better motivate CocoRNA's sample efficiency and to build a better connection with HER, SAER should have experimented on a sparse reward environment.

9. The empirical evaluation is done against a limited amount (and type) of baselines. The proposed method is compared only against RL-based methods. The paper only cites antaRNA [1] and MCTS-RNA [2] approaches, however, these search-based fundamental approaches should be included as baselines; which would support CocoRNA's performance against a diverse set of baselines. Furthermore, another valid baseline from literature that combines RL with local search [3] could have been considered.

10. It would have supported the generalizability of the proposed approach better if an empirical analysis on any other biological sequence design domain such as protein design (which has some benchmark datasets e.g. GB1) was presented.

11. For a thorough analysis, training performance could have been provided. Further, an ablation over the trained policy (e.g. under different training steps or hyperparameters) could have demonstrated the effectiveness of CocoRNA under various training settings.

12. There is no clear discussion on the limitations and potential drawbacks of the method.

**Minor:**

- In line 114, the reference (Eastman et al…) is doubled. \cite{} should be corrected to avoid repetition.

- Regarding notations, instead of using the notation k for several places, another symbol could be used. Specifically, it is used both in equation (5) to denote the subsequence of nucleotides and in equations (7), (9) to denote cumulative future rewards.

> [1] Bal, M. I., Sessa, P. G., Mutny, M., & Krause, A. (2023). Optimistic Games for Combinatorial Bayesian Optimization with Applications to Protein Design. NeurIPS 2023 Workshop on Adaptive Experimental Design and Active Learning in the Real World. https://openreview.net/forum?id=ScOvmGz4xH (or alternatively: http://arxiv.org/abs/2409.18582)

> [2] Kleinkauf, R., Houwaart, T., Backofen, R., & Mann, M. (2015). antaRNA–Multi-objective inverse folding of pseudoknot RNA using ant-colony optimization. BMC bioinformatics, 16, 1-7.

> [3] Yang, X., Yoshizoe, K., Taneda, A., & Tsuda, K. (2017). RNA inverse folding using Monte Carlo tree search. BMC bioinformatics, 18, 1-12.

> [4] Eastman, P., Shi, J., Ramsundar, B., & Pande, V. S. (2018). Solving the RNA design problem with reinforcement learning. PLoS computational biology, 14(6), e1006176.

> [5] Angermueller, C., Dohan, D., Belanger, D., Deshpande, R., Murphy, K., & Colwell, L. (2019). Model-based reinforcement learning for biological sequence design. In International conference on learning representations. (ICLR)

> [6] Feng, L., Nouri, P., Muni, A., Bengio, Y., & Bacon, P. L. (2022). Designing biological sequences via meta-reinforcement learning and bayesian optimization. arXiv preprint arXiv:2209.06259.

**Questions:**

1. Why the proposed method is only applied to RNA design? Can the method be extended to any other biological sequence (DNA, protein, peptide) design? If yes, why restrict its applicability/generalization?

2.  As also stated in the first weakness, what is the main source of benefit of using an RL method, instead of performing collective design for combinatorial optimization as in [1]?

3. How does CocoRNA deal with sparse and/or delayed rewards?

4. How does CocoRNA generalize across different target structures?

5. How is SAER related to/inspired by HER? How the goals are sampled in SAER? How do you gradually reduce the SAER operation, this is not clear in the paper.

6. Can you provide some insight/discussion on how embedding SAER into the training process would not yield a suboptimal solution?

7. In Section 5.3 (and Figure 4), the two ablation studies are done on which dataset? Or is it averaged over replications or datasets-- what do error bars represent? It could have been clarified in the figure caption.

---

> ### Author Response · Authors · 2024-11-22
> **Response to Reviewer MViW (Part 1)**
>
> We thank the reviewer for the insightful and useful feedback, please see the following for our response.
>
> **[W1 and Q2]** **Key differences and advantages of CocoRNA over [1]? What is the main source of benefit of using a MARL method？**
>
> First of all, we appreciate the reviewer's suggestion on this new reference [1], which we were not aware when preparing this ICLR submission. However, we respectfully argue that our work is fundamentally different from [1].
>
> - First, [1] is Bayesian optimization (BO) designed to search for the optimal solution in a per-instance manner. In other words, if the problem instance changes, BO needs to start from scratch. In contrast, we go with reinforcement learning (RL) which aims to learn a generalizable problem-solving policy for a class of problems. Once the policy is learned, agents can make decisions based on input information (the specific problem) without further training or optimization.
> - Another key difference is that [1] establishes a game between different optimization variables to decouple the combinatorial decision space into individual decision sets. Our method, on the other hand, trains a set of policies that can cooperatively make dynamic decisions to complete the design task. We are not searching for a solution (a set of optimal variables) but rather developing a collaborative approach to problem-solving.
> - Given the above justification, we think traditional optimization or search-based methods, which require performing online search for every new problem instance, may not be computationally efficient. In contrast, RL, which adaptively make decisions without additional training or optimization, can achieve few-shot or even zero-shot design when encountering new instances. Therefore, we believe this direction will be valuable for life science problems where obtaining new experimental samples are inevitably expensive, if it is not impossible.
> - Due to the complexity of the RNA inverse design problem, previous RL-based works [4,7] have not truly achieved the zero-shot design. For example, [7] require hundreds or even thousands of learning episodes when facing new RNA structures (see **Figure 2**). In contrast, we introduce the MARL framework that decomposes the problem into sub-tasks handled by multiple agents. Combined with other components, CocoRNA not only achieves significantly better performance but also realizes zero-shot design on new instances. Specifically, as shown in **Figure 2** and **Table 3**, CocoRNA can successfully design most new structures in a single episode without any further training or sampling.
> - As we discussed in **Section 2.1**（page 3）, similar ideas appear in [8], and also in [1] as the reviewer pointed out. However, as mentioned in above points, our approach is fundamentally different from both [1] and [8]. Moreover, our theoretical analysis (see line 780, **Appendix B**) support our assertion that CocoRNA enables multiple agents to act cohesively as a group rather than independently, achieving better collective design performance.
>
> Nevertheless, we have added a discussion on [1] in the Related Works section and clarify the key differences in the revised manuscript (see **Section 2.1**).
>
> > [1] Bal, M. I., Sessa, P. G., Mutny, M., & Krause, A. (2023). Optimistic Games for Combinatorial Bayesian Optimization with Applications to Protein Design. NeurIPS 2023 Workshop on Adaptive Experimental Design and Active Learning in the Real World.
>
> > [4] Eastman, P., Shi, J., Ramsundar, B., & Pande, V. S. (2018). Solving the RNA design problem with reinforcement learning. PLoS computational biology, 14(6), e1006176.
>
> > [7] F. Runge, D. Stoll, S. Falkner, and F. Hutter. (2019). Learning to design RNA. ICLR 2019.
>
> > [8] Mirela Andronescu, Anthony P Fejes, Frank Hutter, Holger H Hoos, and Anne Condon. (2004) A new algorithm for RNA secondary structure design. Journal of molecular biology, 336(3):607–624.

---

> ### Author Response · Authors · 2024-11-22
> **Response to Reviewer MViW (Part 2)**
>
> **[W2] A theoretical analysis or some clear discussion/intuition about convergence. Otherwise modify the claims.**
>
> We thank the reviewer's suggestion and we address your concerns from three aspects.
>
> - We have provided a theoretical analysis in **Appendix B** (please see line 780) of the revised manuscript. In summary, the centralized Critic network in our framework possesses global information to learn accurate global value estimates, which in turn guide the distributed agent policies to improve towards optimizing the global objective.
> - Our theoretical results demonstrate, under certain conditions, CocoRNA converges to a local optimum of the multi-agent joint objective function $J$. While gradient descent is inherently a local optimization algorithm, well-designed RL algorithms can enhance exploration to escape poor local optima, increasing the likelihood of finding better solutions. Specifically, PPO incorporates an entropy term in the policy optimization objective, which encourages exploration and helps prevent premature convergence to suboptimal policies.
> - To clarify, our intent is to convey that the centralized Critic can direct the distributed policies towards optimizing the joint value function, rather than each agent optimizing its own local value function independently. Considering that the term "global optimization" in the paper might cause ambiguity, we have revised it to "joint optimization".
>
> **[W3 and Q3]. It would support the empirical performance of CocoRNA better if an analysis of the performance of CocoRNA under sparse and delayed rewards is presented.**
>
> We thank the reviewer for the suggestion. We have conducted ablation experiments under two different delayed/sparse reward settings (see line 1015, **Appendix E.3**).
>
> - First of all, we would like to emphasize that the proposed SAER method is not specifically designed for sparse rewards. We have discussed this aspect in detail in our response to **W8/Q5/Q6**.
> - Nonetheless, to evaluate the robustness of CocoRNA under different reward configurations, we have tested its performance with delayed rewards and terminal rewards and conducted ablation studies by removing the SAER method (**Appendix E.3**).
> - In summary, we find that the delayed reward setting has almost no impact on CocoRNA's performance. However, in the case of terminal rewards, the algorithm experiences a slight performance loss due to the sparse feedback, but it still clearly outperforms the baselines. Under both reward settings, CocoRNA-ablated performs worse than the full CocoRNA, demonstrating that the introduction of SAER helps improve policy learning, particularly under challenging reward configurations.
>
> **[W4 and Q4]. The paper does not present how CocoRNA generalizes across different target structures.**
>
> We respectfully think this is a misunderstanding and we address your concerns from two aspects.
>
> - In our experiments (see **Table 1**), we employed separate training and testing datasets to ensure that the target structures used during testing were entirely unseen during the training phase. Specifically, the training dataset contains $60,000$ different target structures while the testing dataset comprises $650$ different ones. The results in **Table 1** demonstrated that CocoRNA successfully designed $97.85$% of the RNA structures in the testing set.
> - Moreover, as per requested by the reviewer, we have conducted additional experiments (see line 1075, **Appendix E.6**) where the dataset was re-split based on structural similarity, ensuring that the structures in the testing set had greater differences from those in the training set. **Table 13** presents the results of training and testing using this re-split dataset, which are consistent with the results in **Table 1**. This further demonstrates the generalization capability of CocoRNA.
>
> **[W5]. I think the related work section (2.1) should be restructured. There should be a section for RL-based methods that are used to design biological sequences such as [4,5]. Instead of providing general MARL works in detail, this section would have provided the flow from RL to MARL within the problem context.**
>
> We thank the reviewer's suggestion. We have re-structured the Related Work section and added a discussion on reference [1]. Also, we have expanded **Section 2.1** (see line 97) to include a subsection on **Learning-based Methods**, where we have discussed existing RL-based approaches (including [4, 5]), their limitations, and the motivation for introducing MARL in our work.

---

> ### Author Response · Authors · 2024-11-22
> **Response to Reviewer MViW (Part 3)**
>
> **[W6]. How does the performance of CocoRNA change with respect to the agent size?**
>
> We thank the reviewer's suggestion. We have conducted further experiments to examine the impact of agent size on the algorithm's performance (see line 1075, **Appendix E.4**). **Table 11** and **Figure 7** show the performance of CocoRNA under four different agent configurations, using 2, 4, 6, and 8 agents, respectively. We observe that using 4 or 6 agents yields the best performance. When the number of agents is too small, the benefits of multi-agent problem decomposition cannot be fully exploited. On the flip side, when the number of agents is too large, the environment becomes more non-stationary from the perspective of each agent due to the increased interactions and dependencies among agents.
>
> **[W7]. An ablation study considering the decomposition scheme such as position + structure assigned to an agent would have shown the proposed method's flexibility regarding decomposition choices.**
>
> We respectfully argue that we have already conducted relevant experiments to evaluate the impact of decomposition schemes. In our experiments (see **Appendix E.1**, **Table 8**), we tested both position- and structure-based decomposition schemes. The experimental results have demonstrated that these two decomposition schemes have only a marginal impact on the overall performance of CocoRNA. This suggests that CocoRNA is robust to the choice of decomposition strategy. We have made this finding more explicit in the revised manuscript (see line 473).
>
> **[W8, Q5 and Q6]. In Search-augmented Experience Replay (SAER), how are the goals defined? How are additional goals for the replay sampled? How do you gradually reduce the SAER operation? Can you provide some insight/discussion on how embedding SAER into the training process would not yield a suboptimal solution?**
>
> We thank the reviewer for these comments and questions. We address them from the following three aspects.
>
> - First, we apologize for the confusion about "$\cdots$ Inspired by the HER $\cdots$". In fact, different from HER, which directly modifies the reward signals in the experience replay to address sparse reward issues by using additional goals, SAER is designed to alleviate the cold-start problem by providing agents with higher-quality training data during the early stages of training.
> - Further, SAER achieves this by performing a limited amount of local search to identify better actions and then replacing old data in the experience replay with these improved samples. Specifically, during the first $10$% of the training process, SAER operations are applied to $25$% of the action selections. This is reduced to $10$% between $10$% and $30$% of the training progress. After $30$% of the training is completed, SAER operations are no longer performed.
> - Last but not least, unlike HER, SAER does not introduce new goals, ensuring that the original training objectives remain unaffected. The experimental results in **Figure 3** (right), **Figure 6**, **Table 9**, and **Table 10** demonstrate the effectiveness of the SAER method in improving policy learning and overall performance.
>
> **[W9]. Why only compare against RL-based methods.**
>
> Currently, our baselines include only RL-based methods for two primary reasons:
>
> - First, both antaRNA [2] and MCTS-RNA [3] are search-based methods, which share the similar problem as of BO in our response to your **W1**. That is to say, they need to search from scratch when encountering a new RNA structure. This indicates that they are significantly less efficient compared to CocoRNA. For instance, search-based methods may take several hours to optimize each RNA structure, whereas our method, CocoRNA, can generate designs in merely seconds.
> - More importantly, the results reported in [7] has already demonstrated the superiority of RL-based methods (LEARNA and Meta-LEARNA) over the search-based baselines (e.g., antaRNA and MCTS-RNA). Therefore, we see LEARNA and Meta-LEARNA stronger baselines in our experiments
>
> > [2] Kleinkauf, R., Houwaart, T., Backofen, R., & Mann, M. (2015). antaRNA–Multi-objective inverse folding of pseudoknot RNA using ant-colony optimization. BMC bioinformatics, 16, 1-7.
>
> > [3] Yang, X., Yoshizoe, K., Taneda, A., & Tsuda, K. (2017). RNA inverse folding using Monte Carlo tree search. BMC bioinformatics, 18, 1-12.
>
> > [7] F. Runge, D. Stoll, S. Falkner, and F. Hutter. (2019). Learning to design RNA. ICLR 2019.

---

> ### Author Response · Authors · 2024-11-22
> **Response to Reviewer MViW (Part 4)**
>
> **[W10 and Q1]. More experiments on broader biological sequence design problems.**
>
> We thank the reviewer for this suggestion and we address your concern from three aspects:
>
> - First of all, as a central dogma of molecular biology, RNA molecules play crucial roles in governing the rule of life. Given specific RNA folded structures, the ability of decoding the corresponding RNA sequences has profound implications (e.g., RNA product like mRNA vaccine and therapeutics design).
> - Moreover, the RNA inverse design problem itself is very complex and relatively under explored in the literature. It requires to capture/learn the base-pairing relationships and potential nucleotides interactions to obtain a desired structure. This will be aggravated by the massive search space. In particular, for a single RNA structure can be as large as $4^{450}$. This significantly exceeds the complexity of tasks like the GB1 benchmark, which involves $20^4$ variants with 4 residue sites.
> - Given these justifications, we believe the RNA inverse design is an ideal arena for validating the capability of CocoRNA in problems with extremely large and complex design space. The success from our experiments give us sufficient confidence to apply CocoRNA for other biological sequence design problems, which will be part of our future works.
>
> **[W11]. A thorough analysis of training performance, and an ablation over the trained policy.**
>
> We thank the reviewer's suggestion and we address your concerns from three aspects.
>
> - Regarding the training performance, we have presented the learning curves of CocoRNA in **Figure 3**. Additionally, we have included a series of ablation studies and displayed the learning curves under different settings in **Appendix E** (see **Figures 6-8**), which show the training dynamics and convergence behavior of our algorithm under various configurations.
> - We found CocoRNA is relatively sensitive to the learning rate. In our main experiments, we use a learning rate of $1 \times 10^{-5}$, which is lower than the settings commonly used in RL algorithms. We believe that high learning rates exacerbate non-stationarity due to abrupt policy updates by agents, making it difficult for them to adapt and learn stable policies. As shown in our experiments (see line 1237, **Appendix E.7**, **Figure 9**), higher learning rates result in highly unstable training processes, confirming this hypothesis.
> - CocoRNA is robust to the other key hyperparameters’ settings (e.g., the discount factor $\gamma$, the GAE parameter $\lambda$, and the PPO clip range $\epsilon$). In our experiments, we adopt standard values commonly used in the literature (see line 946, **Appendix D**).
>
> **[W12]. Limitation discussion**
>
> We discuss limitations as follows.
>
> - First, it is non-trivial to design an accurate and reliable reward model given the complexity of biological systems. We may resort to large-scale pretrained models (e.g., RNA/protein language models). However, these models are often too large and thus are computationally expensive. One potential solution is to go with a model-based RL, where an explicit model of the environment is learned and used to predict future states and rewards more efficiently.
> - Another potential limitation is the decomposition method considered in the current CocoRNA. Given the high-dimensional nucleotide design space, it will be more promising to study an adaptive decomposition mechanism. This might be achieved by designing hierarchical policies, where high-level agents make decisions about task decomposition and low-level agents focus on specific sub-tasks.
> - Last but not least, this paper chooses the RNA inverse design problem as a proof-of-concept study. However, we believe CocoRNA or even MARL will be a potent method for solving such highly complex structured space problems. As part of our next step, we will explore other biological sequence design problems such as protein.
>
> We have added the relevant discussion into the updated manuscript (see line 1344, **Appendix F.2**).
>
> **[Q7]. In Section 5.3 (and Figure 4), the two ablation studies are done on which dataset? Or is it averaged over replications or datasets-- what do error bars represent?**
>
> We apologize for causing the confusion. Our ablation studies were conducted using the Rfam dataset. Each experiment was performed over $6$ independent runs with different random seeds. The light-colored parts in **Figure 3** (original Figure 4) represent the standard deviation. We have clarified these in the revised figure caption (see line 508).
>
> **[Minor 1, 2]. In line 114, the reference (Eastman et al…) is doubled. Regarding notations, k is used both in equation (5) to denote the subsequence of nucleotides and in equations (7), (9) to denote cumulative future rewards.**
>
> We apologize for these issues. We have corrected the duplicated reference (please see line 100), and have replaced the symbol $k$ in Equation (5) with $\kappa$ (see line 248).

---

> ### Author Response · Authors · 2024-11-26
> **Follow-up message to Reviewer MViW**
>
> Dear Reviewer MViW,
>
> Sorry for chasing the rebuttal. We thank you very much for your detailed and constructive suggestions and comments to our work. Would you please kindly let us know whether our responses really address your concerns? Or if it is the other way around, please feel free to let us know your further concerns and we are enthusiastic in this discussion.
>
> Thank you very much!

---

> > ### Comment · Reviewer_MViW · 2024-11-26
> > **Thank you for your response; I have additional questions**
> >
> > Thanks a lot to the authors, with the addition of ablation studies and discussions on the updated manuscript. I have these additional questions and observations about your response:
> >
> > - **Q:** It has been shown that many CTDE approaches would get stuck in some local minima and thus lose their optimality
> > guarantee even in toy tabular tasks. CocoRNA guarantees local optimality. Local approximations are of course valuable too, however, I am wondering if the authors could provide some insight on global optimality, or what might help CocoRNA to go beyond local optima? I think not guaranteeing the global optimum policy should be added in the Limitations section as well.
> > - **[W7]:** What I meant by the decomposition scheme of position + structure is that what if e.g. an agent is specifically set to be responsible from the first position *and* the dot structure? As far as I understood from Appendix E1, you have treated these two decompositions separately. That is why, I stated it as this is a more specific decomposition than the structure-only decomposition.
> > - **[W9]:** I appreciate the discussion on baseline choices, however, I still believe that showing CocoRNA's effectiveness against a diverse set of baselines that address the same problem is important, particularly since the method is proposed for the RNA secondary structure design problem. Efficiency is the key advantage of CocoRNA, however, will the generated designs be better than those search or optimization-based methods? Since all of them provide a local approximation, CocoRNA too. This would have quantitatively supported how CocoRNA is effective against a baseline that does not rely on a reliable reward model.

---

> > > ### Author Response · Authors · 2024-11-30
> > > **Response to additional questions of Reviewer MViW (Part 2)**
> > >
> > > **[W7-2] What I meant by the decomposition scheme of position + structure is that what if e.g. an agent is specifically set to be responsible from the first position *and* the dot structure? As far as I understood from Appendix E1, you have treated these two decompositions separately. That is why, I stated it as this is a more specific decomposition than the structure-only decomposition.**
> > >
> > > We apologize for misunderstanding the reviewer’s comment before. From our understanding, if an agent is set to be responsible for a specific position and a specific structure as suggested by the reviewer, it means that each agent is assigned to a specific nucleotide position in the RNA sequence. We respectfully argue that this method is not applicable to CocoRNA for the following two reasons.
> > >
> > > - First, our goal is to train a generalizable RNA design policy that can be applied to RNA structures of varying lengths and complexities. Assigning an agent to each specific position would mean that the number of agents varies with the length of the RNA sequence. This would require retraining the policy for each new RNA structure, which is precisely what we aim to avoid.
> > > - Second, if we assign an agent to each nucleotide position, longer RNA sequences would require hundreds of agents. Our experiments (see **Appendix E.4**) show that using too many agents adversely affects performance. Specifically, an excessive number of agents significantly increases environmental non-stationarity, making it harder for agents to learn effective policies due to the constantly changing dynamics introduced by the actions of numerous agents. Our work does not consider large-scale multi-agent problems because using a large number of agents is unnecessary and counterproductive in this context.
> > >
> > > **[W9-2] Efficiency is the key advantage of CocoRNA, however, will the generated designs be better than those search or optimization-based methods? This would have quantitatively supported how CocoRNA is effective against a baseline that does not rely on a reliable reward model.**
> > >
> > > We thank the reviewer's suggestion and we address your concerns from the following two aspects.
> > >
> > > - For search-based methods, based on the experimental results from LEARNA [7], the performance of antaRNA [2] is better than MCTS-RNA [3]. Therefore, during this rebuttal period, we have added experiments using antaRNA on the Rfam dataset. Due to the time-consuming nature of online searches, we limited the test set to 100 RNA structures (randomly sample $100$ structures from the original test set). We evaluated antaRNA under various computational time limits per RNA structure: 60 seconds, 600 seconds, and 1200 seconds. For the experiments with time limits of 60 seconds and 600 seconds, each RNA structure was designed only once. For the 1200-second time limit, we tested two settings: designing each RNA structure 5 times and 15 times, respectively, and selected the best result among the attempts. We report the average structural distance and the number of fully solved RNA structures (i.e., sequences that fold exactly into the target secondary structures). The results are presented in **Table 14** (see line 1264, **Appendix E.8**).
> > >
> > >   As shown in Table 14, although antaRNA achieves relatively low average structural distances (only a few nucleotides do not match the target structures), the number of RNA structures it fully solves ($22$%) is significantly lower than that achieved by CocoRNA ($97.85$%, **Table 1**). This may be due to the presence of longer and more complex structures in the dataset, which affect optimization efficiency and make antaRNA more prone to getting stuck in local optima. Moreover, the maximum time limit of 1200 seconds per structure is substantially higher than the computational time required by CocoRNA. This supports the advantage of our method.
> > >
> > > - Moreover, we would like to clarify that while search-based methods like antaRNA do not require a reward signal in the same way as RL methods, they still rely on calling the RNA folding algorithm to compute the objective function for evaluating candidate solutions. Both methods require a reliable model to evaluate the fitness of solutions to guide the search or learning process.

---

> > > ### Author Response · Authors · 2024-12-02
> > > **Follow-up message to Reviewer MViW**
> > >
> > > Dear Reviewer MViW,
> > >
> > > Sorry for chasing this because the rebuttal window is very approaching the end. We sincerely appreciate the time and effort you have dedicated to reviewing our submission. We hope our responses have fully addressed your concerns. If you have any further questions or if there’s any additional information we can provide to assist in your evaluation, please do not hesitate to let us know.
> > >
> > > Thank you very much for your valuable time!

---

> ### Author Response · Authors · 2024-11-30
> **Response to additional questions of Reviewer MViW (Part 1)**
>
> **[Q8]** **I am wondering if the authors could provide some insight on global optimality, or what might help CocoRNA to go beyond local optima? I think not guaranteeing the global optimum policy should be added in the Limitations section as well.**
>
> We thank the reviewer for this further question about the convergence property of CocoRNA. We respectfully think the guarantee of global optimal policy is not an issue of CTDE methods. Our justifications are the following five aspects.
>
> - Firstly, in theory, gradient-based deep learning and reinforcement learning (RL) methods often does not guarantee the global optimality but a convergence to local optima, due to the non-convex nature of the optimization landscape. However, from our experience and some studies in the literature (e.g., [11-13]), it has been accepted that the ability of RL algorithms to go beyond local optima and approach global optima depends significantly on how well they balance exploration and exploitation.
> - The concern about CTDE approaches getting stuck in local minima often arises with naive reinforcement learning algorithms, such as the original Actor-Critic methods. However, recent advanced multi-argent RL (MARL) algorithms have demonstrated strong performance in complex tasks [9,10]. Specifically, CocoRNA is built upon PPO, which incorporates several mechanisms to enhance exploration:
>     - The policy network outputs a probability distribution over actions, and actions are sampled from this distribution during training. This stochasticity encourages exploration by allowing agents to try different actions, even those with lower estimated value, which helps in escaping local optima.
>     - PPO includes an entropy bonus in the objective function, which explicitly encourages the policy to maintain high entropy. This prevents the policy from becoming too deterministic too quickly, avoiding premature convergence to suboptimal policies and promoting continued exploration.
> - Moreover, in the context of the RNA design task, the multi-agent framework of CocoRNA further aids in overcoming local optima. Unlike single-agent methods that mutate one nucleotide at one step, our method allows multiple agents to act simultaneously on different parts of the RNA sequence. This introduces greater diversity and increases the exploration of the state space, thereby increasing the opportunities to discover better policies.
> - In addition, we would like to use our empirical results to support our justifications. As the results shown in **Table 1** (line 379) and **Table 14** (line 1264, **Appendix E.8**), classic optimization methods such as antaRNA can only solve $22$% of RNA structures while the policy learned by CocoRNA can solve $97.85$%. This not only demonstrates that there indeed exists many local optima in the RNA design task, it also gives us confidence that CocoRNA has a good capability to avoid getting stuck at poor local optima.
> - Last but not least, we thank the reviewer’s suggestion on discussing the convergence property. In our revised manuscript, we have added a discussion about the theoretical limitations on the convergence property in **Appendix B.2** (line 844). Furthermore, we will rephrase the Limitation section and move it from the current **Appendix F.2** into the main manuscript in the camera-ready version, and we will add another limitation discussion about not guaranteeing the global optimum policy.
>
> > [9] Yu, C., Velu, A., Vinitsky, E., Gao, J., Wang, Y., Bayen, A., & Wu, Y. (2022). The surprising effectiveness of ppo in cooperative multi-agent games. Advances in Neural Information Processing Systems, 35, 24611-24624.
>
> > [10] Rashid, T., Samvelyan, M., De Witt, C. S., Farquhar, G., Foerster, J., & Whiteson, S. (2020). Monotonic value function factorisation for deep multi-agent reinforcement learning. Journal of Machine Learning Research, 21(178), 1-51.
>
> > [11] Sutton, R. S. (2018). Reinforcement learning: An introduction.
>
> > [12] Osband, I., Blundell, C., Pritzel, A., & Van Roy, B. (2016). Deep exploration via bootstrapped DQN. Advances in Neural Information Processing Systems, 29, 4026-4034.
>
> > [13] Jin, C., Yang, Z., Wang, Z. & Jordan, M.I. (2020). Provably efficient reinforcement learning with linear function approximation. Proceedings of Thirty Third Conference on Learning Theory, 2137-2143.

---

### Meta-Review · Area_Chair_3xVp · 2024-12-21

**Metareview:**

This work proposes a cooperative RL method for RNA secondary structure design. This approach splits the RNA design problem into subtasks which are then given to manny different agents to solve them collaboratively. Unfortunately the general reviewing consensus was that the proposed work as is does not present sufficiently distinct contributions from related work to argue an independent case. Other reproducibility issues were raised.

**Additional Comments On Reviewer Discussion:**

The main concerns raised by reviewers were
1. Relationship with related work.
2. Reproducibility.There is no accessible code to reproduce the claimed results.
3. Lack of theoretical analysis or other convergence results.

---

### Decision · Program_Chairs · 2025-01-22

Reject